# PARTCO: PART-LEVEL CORRESPONDENCE PRIORS ENHANCE CATEGORY DISCOVERY

## ABSTRACT

Generalized Category Discovery (GCD) aims to identify both known and novel categories within unlabeled data by leveraging a set of labeled examples from known categories. Existing GCD methods primarily depend on semantic labels and global image representations, often overlooking the detailed part-level cues that are crucial for distinguishing closely related categories. In this paper, we introduce PartCo, short for Part-Level Correspondence Prior, a novel framework that enhances category discovery by incorporating part-level visual feature correspondences. By leveraging part-level relationships, PartCo captures finer-grained semantic structures, enabling a more nuanced understanding of category relationships. Importantly, PartCo seamlessly integrates with existing GCD methods without requiring significant modifications. Our extensive experiments on multiple benchmark datasets demonstrate that PartCo significantly improves the performance of current GCD approaches, achieving state-of-the-art results by bridging the gap between semantic labels and part-level visual compositions, thereby setting new benchmarks for GCD. Code will be made publicly available.

## 1 INTRODUCTION

Supervised deep learning models have fundamentally transformed computer vision, showcasing exceptional proficiency in classifying predefined image categories. Models trained on extensive labeled datasets achieve high accuracy and robustness in distinguishing known classes within controlled environments. However, their performance significantly diminishes when confronted with samples from categories that were neither present nor represented during training. This limitation impedes the deployment of intelligent systems in dynamic, real-world scenarios where encountering previously unseen categories is inevitable. To address this challenge, Generalized Category Discovery (GCD) (Vaze et al., 2022a) has emerged as a pivotal task. As depicted in Fig. 1, GCD aims to automatically identify and categorize both known and novel classes within unlabeled data by leveraging a modest set of labeled examples from known categories. Unlike traditional supervised learning, which operates within a rigid framework of predefined categories, GCD extends the model's capability to recognize and incorporate novel, unseen categories alongside known ones.

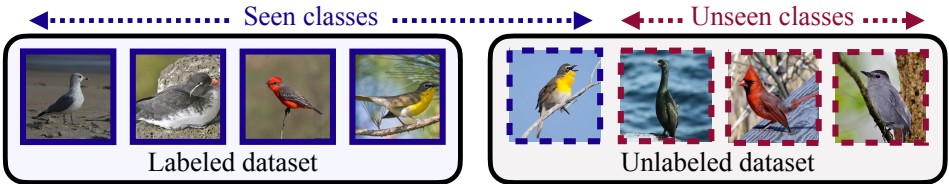

Figure 1: **Generalized Category Discovery:** Given a labeled subset contains seen classes, the task is to categorize the unlabeled images, which may belong to seen or unseen classes.

A growing body of literature in GCD emphasizes the significance of object parts as effective conduits for transferring knowledge between "seen" and "unseen" categories (Vaze et al., 2022a; Wang et al., 2024). Object parts encapsulate fine-grained visual features that are often shared across different categories, facilitating the generalization to novel classes.

However, recent approaches predominantly rely on global representations derived from the classification token of transformer-based models. While these global features capture the overall semantic content of an image, they inherently abstract away detailed part-level information, which is vital for distinguishing closely related categories. For instance, Wang et al. (2024) introduces a spatial prompt tuning method that learns pixel-level prompts around local image regions to incorporate part-level information. Although innovative, this method does not account for the inherent variability in object parts, such as differing scales, orientations, or varying numbers of parts due to occlusions (Fig. 2). This motivates an explicit part-aware prior that is robust to scale, pose, and occlusion.

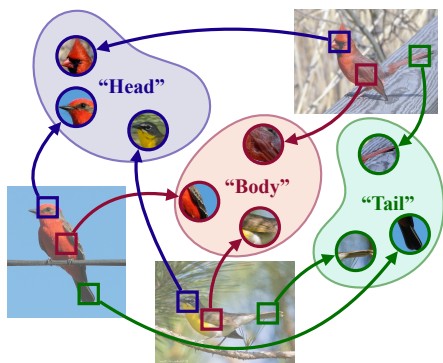

Figure 2: **Part variability.** Parts (head, body, tail) vary in scale, pose, and visibility yet still correspond across images, motivating part-aware priors beyond global features.

Vision Transformer (ViT) models, in addition to the classification token, incorporate patch tokens that encapsulate high-dimensional features for each image patch. These patch tokens inherently contain part-level observations, offering a granular perspective of the image's composition. However, directly utilizing these patch tokens presents several challenges: the absence of explicit part-level information, the presence of foreground-background noise, and varying object scales and orientations across samples. These issues necessitate an effective *supervisory signal* to fully harness the potential of patch token representations in ViT models.

Recent advancements in self-supervised vision foundation models, particularly the ViT-based DINO variants (Caron et al., 2021; Oquab et al., 2024; Siméoni et al., 2025), have demonstrated remarkable generalizability across various tasks. These models excel at extracting high-dimensional patch token features that capture detailed and localized semantic information for each image patch. Unlike the classification token, which summarizes the entire image, patch tokens focus on specific parts, providing a more detailed view of the image's composition. Importantly, these enhanced feature descriptors inherently provide the necessary part-level correspondence labels, serving as an ideal supervisory signal for leveraging patch tokens within the GCD framework.

To address these challenges, we propose *PartCo*, short for **Part**-Level **Co**rrespondence Prior, a versatile framework designed to introduce part-level correspondence labels into the GCD process. By explicitly guiding ViT patch token features with these correspondence labels, PartCo better leverages the utilization of the model's rich feature representations. Additionally, we introduce a novel part-level correspondence loss that effectively leverages these part-level features, ensuring that detailed object part information is accurately captured and utilized for category discovery. Through comprehensive evaluations on both fine-grained and generic benchmark datasets, PartCo achieves state-of-the-art (SOTA) performance, setting a new standard for the GCD task.

## 2 PRELIMINARIES

**Problem statement.** Generalized Category Discovery (GCD) aims to develop a model that accurately classifies unlabeled samples from known categories while simultaneously clustering those from novel, unseen categories. Consider an unlabeled dataset $\mathbf{D}_u = \{(\mathbf{x}_i^u, y_i^u)\} \subset \mathbf{X} \times \mathbf{Y}_u$ and a labeled dataset $\mathbf{D}_l = \{(\mathbf{x}_i^l, y_i^l)\} \subset \mathbf{X} \times \mathbf{Y}_l$, where $\mathbf{Y}_u$ and $\mathbf{Y}_l$ represent the label sets for unlabeled and labeled data, respectively. The unlabeled dataset contains samples from both known categories (included in $\mathbf{Y}_l$) and unknown categories, specifically $\mathbf{Y}_l \subset \mathbf{Y}_u$. Let $M = |\mathbf{Y}_l|$ denote the number of labeled categories. We assume the total number of categories, $K = |\mathbf{Y}_l \cup \mathbf{Y}_u|$, is known, as established in prior studies (Han et al., 2021; Vaze et al., 2023). When this information is unavailable, methods such as those in Han et al. (2019); Vaze et al. (2022a) can provide reliable estimates.

**Baselines.** The *non-parametric* baseline (Vaze et al., 2022a; Rastegar et al., 2024) for GCD is introduced by fine-tuning the pretrained DINO model (Caron et al., 2021; Dosovitskiy et al., 2021). The loss function generally integrates both self-supervised and supervised contrastive losses. For two

augmented views $\mathbf{x}_i$ and $\mathbf{x}'_i$ over a mini-batch $B$, we obtain $\ell_2$-normalized features $\mathbf{z}_i = \psi(\phi(\mathbf{x}_i))$ and $\mathbf{z}'_i = \psi(\phi(\mathbf{x}'_i))$, where $\phi$ is the backbone and $\psi$ is the projection head; $\tau_r$ denotes the temperature parameter. The contrastive losses are then defined as:

$$\mathcal{L}^u_{rep} = \frac{1}{|B|} \sum_{i \in B} - \log \frac{\exp(\mathbf{z}_i \cdot \mathbf{z}'_i / \tau_r)}{\sum_{j \neq i} \exp(\mathbf{z}_i \cdot \mathbf{z}'_j / \tau_r)}, \quad \mathcal{L}^s_{rep} = \frac{1}{|B_l|} \sum_{i \in B_l} \frac{1}{|\mathbb{N}_i|} \sum_{q \in \mathbb{N}_i} - \log \frac{\exp(\mathbf{z}_i \cdot \mathbf{z}_q / \tau_r)}{\sum_{j \neq i} \exp(\mathbf{z}_i \cdot \mathbf{z}_j / \tau_r)}. \tag{1}$$

Here, $\mathbb{N}_i$ contains indices of labeled samples sharing the same label $y^l_i$ as $\mathbf{x}_i$. The total representation loss $\mathcal{L}_{rep}$ is a weighted combination:

$$\mathcal{L}_{rep} = (1 - \lambda_b)\mathcal{L}^u_{rep} + \lambda_b \mathcal{L}^s_{rep}, \tag{2}$$

where $\lambda_b$ is the balancing factor.

The *parametric* baseline from Wen et al. (2023) employs a parametric classifier within a self-distillation framework (Caron et al., 2021). Initialized with $K$ normalized category prototypes $\mathbf{L} = \{\mathbf{l}_1, \ldots, \mathbf{l}_K\}$, the classifier computes the probability for category $k$ as:

$$\mathbf{p}_i^{(k)} = \frac{\exp(\mathbf{o}_i \cdot \mathbf{l}_k / \tau_s)}{\sum_{j=1}^{K} \exp(\mathbf{o_i} \cdot \mathbf{l}_j / \tau_s)}, \tag{3}$$

where $\mathbf{o}_i = \phi(\mathbf{x}_i)/\|\phi(\mathbf{x}_i)\|$ and $\tau_s$ is the student temperature. Soft labels $\mathbf{q}_i$ are generated by a teacher network with temperature $\tau_t$. The unsupervised classification loss $\mathcal{L}^u_{cls}$ is defined as $\mathcal{L}^u_{cls} = \frac{1}{|B|} \sum_{i \in B} \ell_{ce}(\mathbf{q}'_i, \mathbf{p}_i) - \xi \mathcal{H}(\overline{\mathbf{p}})$, where $\overline{\mathbf{p}} = \frac{1}{2|B|} \sum_{i \in B}(\mathbf{p}_i + \mathbf{p}'_i)$ denotes the mean prediction across the mini-batch, $\ell_{ce}$ is the cross-entropy loss and $\mathcal{H}$ is the mean entropy, weighted by $\xi$. For labeled samples, the supervised loss $\mathcal{L}^s_{cls} = \frac{1}{|B_l|} \sum_{i \in B_l} \ell_{ce}(\mathbf{p}_i, \mathbf{y}_i)$ is used. The overall classification loss combines unsupervised and supervised components as: $\mathcal{L}_{cls} = (1 - \lambda_b)\mathcal{L}^u_{cls} + \lambda_b \mathcal{L}^s_{cls}$. Finally, integrating with the non-parametric representation loss in Eq. 2 yields the comprehensive GCD objective:

$$\mathcal{L}_{\text{gcd}} = \mathcal{L}_{cls} + \mathcal{L}_{rep}. \tag{4}$$

**Limitation of baselines.** Although the non-parametric and parametric baselines obtain encouraging results on GCD, they exhibit significant limitations. Primarily, these methods rely solely on the foundation model's classification token (`[CLS]`) representation, which captures only global information about the input data. This exclusive dependence on global representations restricts the models from leveraging part-level or localized information that is essential for distinguishing fine-grained categories. Without incorporating detailed, part-specific features, the baselines may overlook subtle patterns and contextual nuances within the data, leading to less effective performance in category discovery.

## 3 PART-LEVEL CORRESPONDENCE PRIOR (PARTCO) FRAMEWORK

Building upon the motivations outlined in the introduction, we introduce *PartCo*, a novel framework meticulously crafted to harness part-level information from ViT's patch tokens for GCD. Unlike traditional approaches that rely solely on global representations provided by the `[CLS]` token, PartCo fully leverages the rich, high-dimensional features embedded within ViT's patch tokens. By generating and utilizing explicit part-level correspondence labels, PartCo effectively bridges the gap between coarse global features and fine-grained local details. Furthermore, this design allows PartCo to be seamlessly integrated into existing GCD methods, enhancing their performance without necessitating significant modifications.

By making use of these part-level correspondence labels, PartCo fully utilizes the vision foundation model beyond just the `[CLS]` token. These labels act as robust supervisory signals, guiding the patch token features to focus on meaningful object parts and mitigating common challenges such as foreground-background noise and variability in object scales and orientations. This guidance enables the full potential of vision foundation models to be realized, ensuring that both global and local feature representations are explicitly integrated into the GCD process. In the subsequent sections, we detail the construction and utilization of part-level correspondence labels within the PartCo framework.

## 3.1 Constructing Part-Level Correspondence Labels

To construct part-level correspondence labels, we employ a two-step process, illustrated in Fig. 3, leveraging the rich feature representations from the frozen DINOv2 model. This approach ensures robust label inference across both labeled and unlabeled samples by first acquiring relevant PCA projections and then assigning labels through $k$-means clustering.

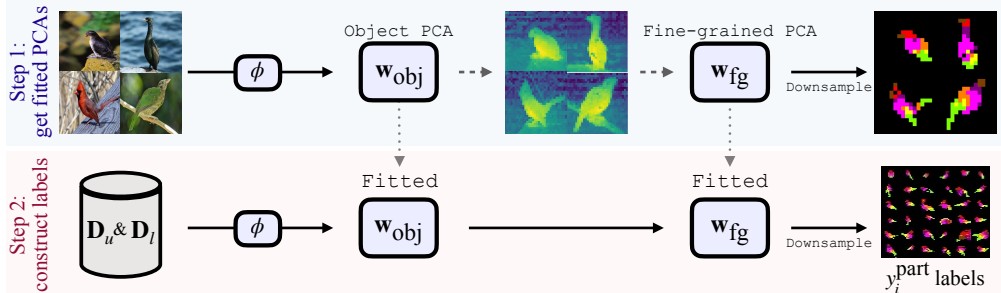

Figure 3: **Overview of part-level correspondence labels construction:** This two-step process begins by applying PCA projections to extract object and detailed features from ViT's patch tokens using a subset of the dataset. These projections are then applied to the entire dataset to generate part-level correspondence labels.

**Step 1: PCA projections.** We begin by sampling a subset of $M$ labeled images from the dataset $\mathbf{D}_l$, ensuring that each sample represents a distinct category. From these images, we extract their patch token features denoted as $\mathbf{F} \in \mathbb{R}^{M \times N \times d}$ using vision foundation model $\phi$, *e.g.*, DINO backbone (Oquab et al., 2024; Siméoni et al., 2025), where $N$ is the number of patch tokens and $d$ is the feature dimension. The first principal component analysis (PCA) is applied to $\mathbf{F}$ to obtain the primary projection vector $\mathbf{w}_{\text{obj}} \in \mathbb{R}^d$, which captures the most significant variation corresponding to object regions: $\mathbf{w}_{\text{obj}} = \text{argmax}_{\mathbf{w}} \frac{\mathbf{w}^\top \mathbf{F}^\top \mathbf{F} \mathbf{w}}{\mathbf{w}^\top \mathbf{w}}$. Using this projection, we compute the objectness score for each patch: $\mathbf{F}_{\text{obj}} = \mathbf{F} \cdot \mathbf{w}_{\text{obj}}$, and generate a binary mask $\mathbf{M}$ by thresholding at $\tau_{\text{obj}} = 0.6$: $\mathbf{M} = \mathbb{1}(\mathbf{F}_{\text{obj}} > \tau_{\text{obj}})$. This mask distinguishes foreground patches from background ones. Subsequently, we perform a second PCA on the masked features to extract fine-grained information. Specifically, we compute the element-wise multiplication $\mathbf{F} \odot \mathbf{M}$ and apply PCA to obtain the projection matrix $\mathbf{w}_{\text{fg}} \in \mathbb{R}^{d \times 3}$, resulting in the fine-grained feature representation: $\mathbf{F}_{\text{fg}} = (\mathbf{F} \odot \mathbf{M}) \cdot \mathbf{w}_{\text{fg}}$. This transformation maps the first three components of the PCA computed over the feature space to RGB.

**Step 2: Label construction.** We determine the optimal number of part-level labels, $k^*$, by applying $k$-means clustering to the normalized fine-grained features $\mathbf{F}_{\text{fg}}$. Each clustering solution is evaluated based on two criteria: (1) *minimum distance* between cluster centers to ensure that the clusters are well separated, reducing overlap and increasing distinctiveness. (2) *balance of cluster sizes*: prevents skewed distributions where some clusters dominate over others, promoting uniformity. We sweep $k$ over a candidate set and select $k^*$ by maximizing $\min_{i \neq j} \|\mathbf{c}_i - \mathbf{c}_j\| \times (\min_i |C_i| / \max_j |C_j|)$, favoring well-separated and balanced clusters. Here, $\mathbf{c}_i$ and $\mathbf{c}_j$ denote the centroids of clusters $i$ and $j$, and $|C_i|$ is the number of samples in cluster $i$. With $k^*$ fixed, we assign part-level correspondence labels to all samples in $\mathbf{D}$. We then define the part label map $y_i^{\text{part}} \in \{1, \ldots, k^*\}$ with resolution following ViT's patch token size as:

$$y_i^{\text{part}} = \arg\min_{\mathbf{c} \in \mathcal{C}} \|\mathbf{F}_{\text{fg}} - \mathbf{c}\|, \tag{5}$$

where $\mathcal{C} = \{\mathbf{c}_1, \mathbf{c}_2, \ldots, \mathbf{c}_{k^*}\}$ represents the set of optimal cluster centers from $k$-means. We refer to these as *1st order part-level correspondence labels*.

**Enhancing granularity in part-level correspondence.** While 1st order part-level labels are adequate for fine-grained datasets due to the presence of shared superclasses, they may be too general for generic datasets lacking such similarities as shown in Fig. 4. To capture more intricate details in these cases, we introduce *2nd order part-level correspondence labels*. This process involves applying an additional PCA on the fine-grained features $\mathbf{F}_{\text{fg}}$ within each 1st order cluster. By doing so, we identify finer distinctions within each part, uncovering common features among similar but distinct parts. This 2nd order of labeling increases the granularity of part-level correspondence, enabling more

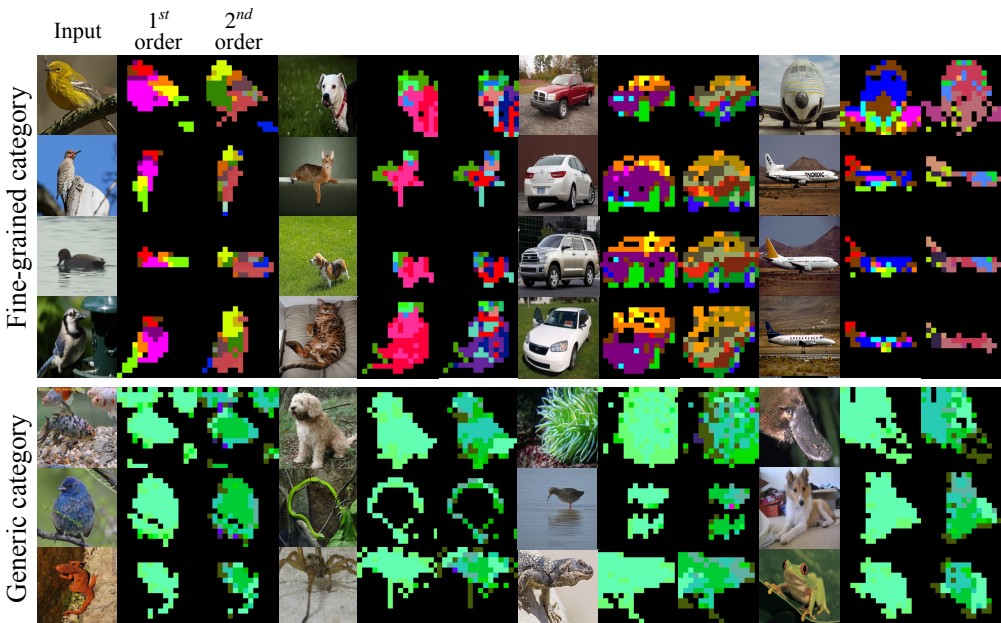

Figure 4: **Visualization of our part-level correspondence labels.** For each image, we generate both first- and second-order labels. First-order labels suffice for fine-grained datasets, while second-order labels capture additional detail for generic datasets. In practice, selecting between 1st- and 2nd-order is straightforward: datasets with subtle, intra-class differences indicate fine-grained samples, whereas datasets with pronounced, inter-class differences indicate generic samples.

precise category discovery in generic datasets where parts exhibit greater diversity and require finer resolution to discern subtle differences. We provide more discussion on our design choice of PCA + DINO patch descriptors and alternatives in Sec. S3.2 of the supplementary.

## 3.2 INTEGRATING PARTCO FRAMEWORK WITH GCD METHOD

After obtaining part-level correspondence labels, as explained in Section 3.1, we incorporate the PartCo framework, illustrated in Fig. 5 (a), into existing GCD methods. This integration is achieved through the introduction of a part-level correspondence loss $\mathcal{L}_{pc}$, which supervises the aggregated part features derived from patch tokens, thereby fostering robust part-level relationships within the ViT's feature representations.

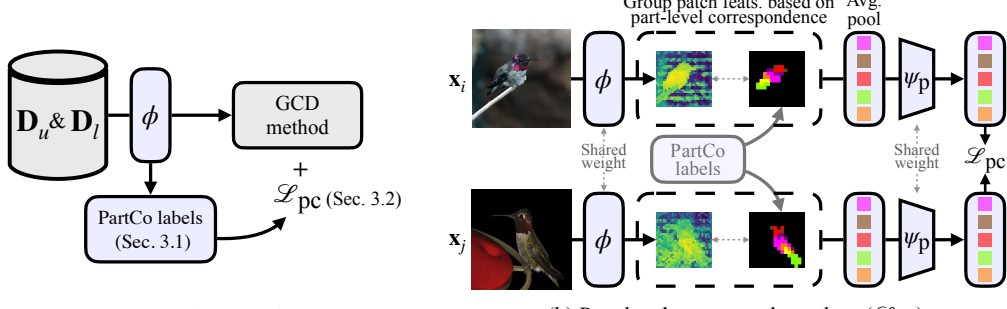

(a) PartCo framework

(b) Part-level correspondence loss ($\mathcal{L}_{pc}$)

Figure 5: **(a) PartCo framework:** Introduces part-level correspondence labels as a plug-and-play module to enhance GCD methods. **(b) Part-level correspondence loss:** Depicts how part-level correspondence loss is integrated into the model to learn relationships between parts in ViT's patch token features.

**Guiding patch token features.** For an input image $\mathbf{x}_i$, we first extract its patch token features using the foundation model $\phi$, yielding $\mathbf{F}_i = \phi(\mathbf{x}_i) \in \mathbb{R}^{N \times d}$. Utilizing the corresponding part-level correspondence labels $y_i^{\text{part}}$, we organize the patch features based on their assigned part categories.

Specifically, for each part category $c \in \mathcal{C}$ (where $\mathcal{C}$ represents the set of all part-level categories), we aggregate the features of patches labeled as $c$ by computing their average: $\mathbf{f}_c = \frac{1}{|\mathcal{P}_c|} \sum_{j \in \mathcal{P}_c} \mathbf{F}_{i,j}$, where $\mathcal{P}_c = \{j \mid y_i^{\text{part}}(j) = c\}$ denotes the set of patch indices corresponding to part category $c$. This pooling operation results in a set of aggregated part-level features $\{\mathbf{f}_c\}_{c \in \mathcal{C}}$, each encapsulating the information of a specific part within the image. To further refine these aggregated features, we employ a part projection head $\psi_p$, which projects each aggregated feature $\mathbf{f}_c$ into a new feature space: $\mathbf{h}_c = \psi_{\text{p}}(\mathbf{f}_c)$, where $\mathbf{h}_c \in \mathbb{R}^{d'}$ represents the projected feature for part category $c$, and $d'$ is the dimensionality of the projected feature space. The projection head $\psi_{\text{p}}$ is typically implemented as a multi-layer perceptron (MLP) that maps the aggregated features to a space optimized for contrastive learning. The overview of the process is shown in Fig. 5 (b).

**Part-level correspondence loss.** To effectively leverage these projected part-level features, we introduce a supervised part contrastive loss $\mathcal{L}_{\text{pc}}^{\text{sup}}$ that operates on the labeled data $\mathbf{D}_l$. This loss encourages features of the same part type and class to be close while separating different parts and/or classes, thereby enhancing the discriminative capability of the model at the part level. Formally, for a batch of $B_l$ labeled samples, the supervised contrastive loss is defined as:

$$\mathcal{L}_{\text{pc}}^{\text{sup}} = \frac{1}{|B_l|} \sum_{i \in B_l} \frac{1}{|\mathcal{C}|} \sum_{c \in \mathcal{C}} \frac{1}{|\mathbb{N}_i^c|} \sum_{q \in \mathbb{N}_i^c} - \log \frac{\exp(\mathbf{h}_c \cdot \mathbf{h}_q / \tau_r)}{\sum_{j \notin \mathbb{N}_i^c} \exp(\mathbf{h}_c \cdot \mathbf{h}_j / \tau_r)}, \quad (6)$$

where $\mathbb{N}_i^c$ contains indices of labeled samples sharing the same label $y_i^l$ and part category $c$ as $\mathbf{x}_i$. This loss function ensures that projected features $\mathbf{h}_c$ of the same part type and category are drawn closer in the feature space, while those of different part type and categories are repelled, thereby fostering more discriminative part-specific representations.

For parametric baselines that incorporate pseudo-labels, in Eq. 3, for unlabeled data $\mathbf{D}_u$, we extend the part-level correspondence loss to include an unsupervised part contrastive loss $\mathcal{L}_{\text{pc}}^{\text{unsup}}$. This loss operates similarly to its supervised counterpart but utilizes pseudo-labels $\mathbf{p}_i$ generated by the model.

$$\mathcal{L}_{\text{pc}}^{\text{unsup}} = \frac{1}{|B_u|} \sum_{i \in B_u} \frac{1}{|\mathcal{C}|} \sum_{c \in \mathcal{C}} \frac{1}{|\mathbb{M}_i^c|} \sum_{q \in \mathbb{M}_i^c} - \log \frac{\exp(\mathbf{h}_c \cdot \mathbf{h}_q / \tau_r)}{\sum_{j \notin \mathbb{M}_i^c} \exp(\mathbf{h}_c \cdot \mathbf{h}_j / \tau_r)}, \quad (7)$$

where $\mathbb{M}_i^c$ contains indices of unlabeled samples sharing the same pseudo label and part category as $\mathbf{x}_i$. This unsupervised loss complements the supervised loss, enabling the model to learn from both labeled and unlabeled data effectively.

**Overall training objective.** The integration of the PartCo framework with existing GCD methods is formalized by combining the GCD's baseline loss in Eq. 4 with the newly introduced part-level correspondence loss. The final training objective is given by:

$$\mathcal{L}_{\text{total}} = \mathcal{L}_{\text{gcd}} + \mathcal{L}_{\text{pc}}, \quad (8)$$

where $\mathcal{L}_{\text{pc}} = (1 - \lambda_b)\mathcal{L}_{\text{pc}}^{\text{unsup}} + \lambda_b \mathcal{L}_{\text{pc}}^{\text{sup}}$ for parametric baselines, or $\mathcal{L}_{\text{pc}} = \lambda_b \mathcal{L}_{\text{pc}}^{\text{sup}}$ for non-parametric ones. By incorporating $\mathcal{L}_{\text{pc}}$, the model utilizes both global features from the `[CLS]` token and detailed part-specific features from the patch tokens. This dual supervision enhances the model's ability to discover and distinguish categories with finer details.

## 4 EXPERIMENTS

In this section, we describe our experimental setups in Sec. 4.1. Next, we present our main results in Sec. 4.2. Finally, in Sec. 4.3 we analyze the effectiveness of PartCo's components and design choices.

### 4.1 EXPERIMENTAL SETUP

**Datasets.** We evaluate our method using several benchmark datasets. Specifically, we use the Semantic Shift Benchmark (SSB) (Vaze et al., 2022b), which includes fine-grained datasets: Caltech-UCSD Birds-200-2011 (CUB) (Wah et al., 2011), Stanford Cars (Krause et al., 2013), and FGVC-Aircraft (Maji et al., 2013). Additionally, we employ generic benchmark datasets: CIFAR10 (Krizhevsky & Hinton, 2009), CIFAR100 (Krizhevsky & Hinton, 2009), and ImageNet-100 (Deng et al., 2009). For each dataset, we utilize the data partitioning strategy specified in (Vaze et al., 2022a). This approach involves selecting a subset of all classes as the known ('Old') classes,

denoted by $\mathbf{Y}_l$. Subsequently, 50% of the images from these known classes are allocated to the labeled dataset $\mathbf{D}_l$, and the remaining images are designated as the unlabeled dataset $\mathbf{D}_u$. Detailed statistics of the datasets are provided in Tab. A, supp. material.

**Evaluation metrics.** We assess the performance of our approach using clustering accuracy (*ACC*; Hungarian-matched), as defined in the existing literature (Vaze et al., 2022a). The *ACC* for the unlabeled dataset $\mathbf{D}_u$ is calculated based on the ground-truth labels $y_i^u$ and the predicted labels $\hat{y}_i^u$ using the following equation: $ACC = \frac{1}{|\mathbf{D}_u|} \sum_{i=1}^{|\mathbf{D}_u|} \mathbb{1}(y_i^u = h(\hat{y}_i^u))$, where $h$ represents the optimal permutation that aligns the predicted cluster assignments with the true labels. Additionally, we report the *ACC* values separately for the 'All', 'Old', and 'New' classes to provide a detailed evaluation of the model's performance across different category groups.

**Implementation details.** We integrate our PartCo framework with the widely used parametric model SimGCD (Wen et al., 2023) and the SOTA non-parametric GCD method SelEx (Rastegar et al., 2024), employing DINO-variants (Oquab et al., 2024; Siméoni et al., 2025) pretrained weights. For SimGCD (Wen et al., 2023), the feature dimension from the backbone $\phi$ is set to 768. The projection head $\psi$, the part projection head $\psi_p$ and the final block of $\phi$ are optimized using the SGD optimizer with an initial learning rate of 0.1, which decays to 0.001 following a cosine annealing schedule, and the balancing factor $\lambda_b$ is fixed at 0.35. Both models are trained for 200 epochs with a batch size of 128. All input images are resized to $224 \times 224$ and augmented to match the DINO pretrained model settings. All results are on a single NVIDIA RTX 4090; PartCo adds no inference cost. The part-level label construction takes around 5–180 min depending on dataset size (Tab. A, supp. material).

**Comparison with other methods.** We compare our method with other representative and SOTA GCD methods: 1) GCD (Vaze et al., 2022a); 2) SimGCD (Wen et al., 2023); 3) $\mu$GCD (Vaze et al., 2023); 4) AMEND (Banerjee et al., 2024); 5) CiPR (Hao et al., 2024); 6) SPTNet (Wang et al., 2024); 7) ProtoGCD (Ma et al., 2025); 8) FlipClass (Lin et al., 2024); 9) SelEx (Rastegar et al., 2024); 10) APL (Dai et al., 2025); 11) AFGCD (Xu et al., 2025); 12) DebGCD (Liu & Han, 2025); 13) PartGCD (Wang et al., 2025a); 14) MOS (Peng et al., 2025); 15) SEAL (He et al., 2025); 16) RLCD (Liu et al., 2025a); 17) AllGCD (Cao et al., 2025); 18) HypCD (Liu et al., 2025b); 19) ConGCD (Tang et al., 2025); and 20) NCGCD (Han et al., 2025). We also report $k$-means clustering results on frozen DINO (Oquab et al., 2024; Siméoni et al., 2025).

## 4.2 EXPERIMENTAL RESULTS

**Benchmark results.** Tab. 1 and 2 report per-dataset results on SSB (fine-grained) and on generic datasets. PartCo consistently improves the parametric model SimGCD and the non-parametric model SelEx, using DINO variants, achieving SOTA results. Unless stated otherwise, all "% gains" below denote *absolute* differences in *ACC*. On SSB, in terms of overall 'All' *ACC* across datasets (Fig. 6), PartCo-SimGCD improves by +9.8% with DINOv2 and +3.8% with DINOv3; PartCo-SelEx improves by +2.4% and +2.9%, respectively. On generic datasets, the gains are +0.3%/+0.9% for PartCo-SimGCD and +2.2%/+1.0% for PartCo-SelEx (v2/v3). On SSB per-dataset results, we

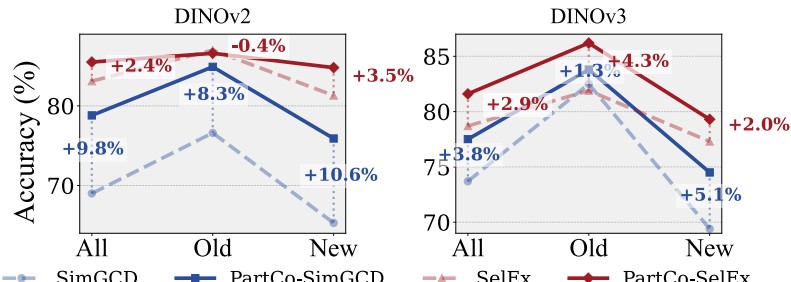

Figure 6: **Average absolute % gain** of PartCo over each baseline on SSB benchmark.

observe large *absolute* % gains across all three datasets: FGVC-Aircraft (DINOv2: +12.6%, DINOv3: +3.9%), CUB (DINOv2: +9.6%, DINOv3: +2.9%), and Stanford-Cars (DINOv2: +7.4%, DINOv3: +4.6%). On generic per-dataset results, gains are smaller but steady. These trends are consistent across DINOv2 and DINOv3, and PartCo integration consistently boosts GCD methods, yielding new state-of-the-art results across diverse datasets.

Table 1: Comparison of GCD methods on the SSB benchmark datasets. Results are reported in *ACC* across the 'All', 'Old' and 'New' categories.

| Method | Venue | Backbone | CUB All | Old | New | Stanford-Cars All | Old | New | FGVC-Aircraft All | Old | New | Average All | Old | New |
|---|---|---|---|---|---|---|---|---|---|---|---|---|---|---|
| k-means | - | DINOv2 | 67.6 | 60.6 | 71.1 | 29.4 | 24.5 | 31.8 | 18.9 | 16.9 | 19.9 | 38.6 | 34.0 | 40.0 |
| GCD | CVPR'22 | DINOv2 | 71.9 | 71.2 | 72.3 | 65.7 | 67.8 | 64.7 | 55.4 | 47.9 | 59.2 | 64.3 | 62.3 | 65.4 |
| $\mu$GCD | NeurIPS'23 | DINOv2 | 74.0 | 75.9 | 73.1 | 76.1 | 91.0 | 68.9 | 66.3 | 68.7 | 65.1 | 72.1 | 78.6 | 69.0 |
| CiPR | TMLR | DINOv2 | 78.3 | 73.4 | 80.8 | 66.7 | 77.0 | 61.8 | - | - | - | - | - | - |
| ProtoGCD | TPAMI | DINOv2 | 75.7 | 81.5 | 72.9 | 77.6 | 90.5 | 71.5 | 71.1 | 76.3 | 68.5 | 74.8 | 82.7 | 71.0 |
| APL | CVPR'25 | DINOv2 | 75.1 | 79.1 | 73.2 | 73.4 | 87.6 | 66.7 | 68.8 | 74.1 | 66.6 | 72.4 | 80.3 | 68.8 |
| AFGCD | ICCV'25 | DINOv2 | 76.5 | 77.3 | 76.0 | 75.5 | 86.2 | 70.3 | 68.1 | 75.9 | 64.1 | 73.4 | 79.8 | 70.1 |
| DebGCD | ICLR'25 | DINOv2 | 77.5 | 80.8 | 75.8 | 75.4 | 87.7 | 69.5 | 71.9 | 76.0 | 69.8 | 74.9 | 81.5 | 71.7 |
| MOS | CVPR'25 | DINOv2 | 81.1 | 82.1 | 80.6 | 75.5 | 89.6 | 68.7 | 69.7 | 78.1 | 65.5 | 75.4 | 83.3 | 71.6 |
| PartGCD | TMM | DINOv2 | 77.6 | 80.6 | 76.1 | 78.2 | 88.7 | 73.1 | 71.1 | 75.6 | 68.8 | 75.6 | 81.6 | 72.7 |
| SEAL | NeurIPS'25 | DINOv2 | 76.7 | 78.3 | 75.9 | 77.7 | 88.7 | 72.4 | 74.6 | 73.2 | 75.3 | 76.3 | 80.1 | 74.5 |
| RLCD | ICML'25 | DINOv2 | 78.7 | 79.5 | 78.3 | 79.5 | 91.8 | 73.5 | 72.6 | 77.3 | 70.3 | 76.9 | 82.9 | 74.0 |
| AllGCD | ICCV'25 | DINOv2 | 78.4 | 82.8 | 76.2 | 76.2 | 88.3 | 70.4 | - | - | - | - | - | - |
| Hyp-SimGCD | CVPR'25 | DINOv2 | 77.6 | 77.9 | 77.4 | 82.5 | 85.8 | 81.0 | 76.4 | 70.3 | 79.4 | 78.8 | 78.0 | 79.3 |
| ConGCD-SelEx | ICCV'25 | DINOv2 | 86.3 | 87.4 | 85.8 | 79.8 | 93.1 | 73.3 | 81.7 | 83.3 | 81.0 | 82.6 | 87.9 | 80.0 |
| NCGCD-SelEx | NeurIPS'25 | DINOv2 | 87.9 | 85.3 | 89.2 | 81.1 | 90.9 | 79.3 | 83.1 | 88.3 | 82.5 | 84.0 | 88.2 | 83.7 |
| Hyp-SelEx | CVPR'25 | DINOv2 | 90.7 | 85.3 | 93.4 | 83.8 | 93.3 | 79.2 | 83.4 | 82.0 | 84.1 | 86.0 | 86.9 | 85.5 |
| SimGCD | ICCV'23 | DINOv2 | 71.5 | 78.1 | 68.3 | 71.5 | 81.9 | 66.6 | 63.9 | 69.9 | 60.9 | 69.0 | 76.6 | 65.3 |
| PartCo-SimGCD | **Ours** | DINOv2 | **81.1** | **82.4** | **80.5** | **78.9** | **91.5** | **72.8** | **76.5** | **80.9** | **74.4** | **78.8** | **84.9** | **75.9** |
| SPTNet | ICLR'24 | DINOv2 | 76.3 | 79.5 | 74.6 | 72.3 | 82.0 | 67.5 | 68.0 | 75.2 | 60.9 | 72.2 | 78.9 | 67.6 |
| PartCo-SPTNet | **Ours** | DINOv2 | **82.6** | **82.3** | **81.8** | **80.1** | **92.0** | **73.5** | **78.6** | **77.6** | **79.0** | **80.4** | **84.0** | **78.1** |
| FlipClass | NeurIPS'24 | DINOv2 | 79.3 | 80.7 | 78.5 | 78.0 | 88.0 | 73.2 | 71.1 | 75.1 | 69.1 | 76.1 | 81.3 | 73.6 |
| PartCo-FlipClass | **Ours** | DINOv2 | **85.2** | **86.3** | **84.7** | **80.5** | **92.7** | **74.6** | **77.1** | **78.5** | **76.4** | **80.9** | **85.8** | **78.6** |
| SelEx | ECCV'24 | DINOv2 | 87.4 | **85.1** | 88.5 | 82.2 | **93.7** | 76.7 | 79.8 | 82.3 | 78.6 | 83.1 | **87.0** | 81.3 |
| PartCo-SelEx | **Ours** | DINOv2 | **90.6** | 84.5 | **93.2** | **82.5** | 91.8 | **78.0** | **83.4** | **83.6** | **83.3** | **85.5** | 86.6 | **84.8** |
| k-means | - | DINOv3 | 69.8 | 64.7 | 72.3 | 59.0 | 50.8 | 63.1 | 40.3 | 35.3 | 42.8 | 56.4 | 50.3 | 59.4 |
| SimGCD | ICCV'23 | DINOv3 | 75.9 | **83.8** | 72.0 | 73.9 | 79.4 | 71.3 | 71.4 | **84.4** | 65.0 | 73.7 | 82.5 | 69.4 |
| PartCo-SimGCD | **Ours** | DINOv3 | **78.8** | 83.4 | **76.6** | **78.5** | **86.5** | **74.6** | **75.3** | 81.6 | **72.2** | **77.5** | **83.8** | **74.5** |
| SelEx | ECCV'24 | DINOv3 | 83.5 | 78.1 | 86.2 | 78.8 | 90.3 | 73.3 | 73.9 | 77.3 | 72.4 | 78.7 | 81.9 | 77.3 |
| PartCo-SelEx | **Ours** | DINOv3 | **86.1** | **84.4** | **87.0** | **81.7** | **92.1** | **76.6** | **76.9** | **82.2** | **74.2** | **81.6** | **86.2** | **79.3** |

Table 2: Comparison of GCD methods on the generic benchmark datasets. Results are reported in *ACC* across the 'All', 'Old' and 'New' categories.

| Method | Venue | Backbone | CIFAR10 All | Old | New | CIFAR100 All | Old | New | ImageNet-100 All | Old | New | Average All | Old | New |
|---|---|---|---|---|---|---|---|---|---|---|---|---|---|---|
| k-means | - | DINOv2 | 94.9 | 95.2 | 94.8 | 70.9 | 70.8 | 72.1 | 78.3 | 80.5 | 77.2 | 81.4 | 82.2 | 81.4 |
| GCD | CVPR'22 | DINOv2 | 97.8 | 99.0 | 97.1 | 79.6 | 84.5 | 69.9 | 78.5 | 89.5 | 73.0 | 85.3 | 91.0 | 80.0 |
| AMEND | WACV'24 | DINOv2 | 97.7 | 96.6 | 98.3 | 83.5 | 83.0 | 84.5 | 87.3 | 95.1 | 83.4 | 89.5 | 91.6 | 88.7 |
| CiPR | TMLR | DINOv2 | 99.0 | 98.7 | 99.2 | 90.3 | 89.0 | 93.1 | 88.2 | 87.6 | 88.5 | 92.5 | 91.8 | 93.6 |
| SPTNet | ICLR'24 | DINOv2 | 98.9 | 99.1 | 98.8 | 89.0 | 91.5 | 79.2 | 90.1 | 96.1 | 87.1 | 92.7 | 95.6 | 88.4 |
| FlipClass | NeurIPS'24 | DINOv2 | 99.0 | 98.2 | 99.4 | 91.7 | 90.4 | 94.2 | 91.0 | 96.3 | 88.3 | 93.9 | 95.0 | 94.0 |
| DebGCD | ICLR'25 | DINOv2 | 98.9 | 97.5 | 99.6 | 90.1 | 90.9 | 88.6 | 93.2 | 97.0 | 91.2 | 94.1 | 95.1 | 93.1 |
| SEAL | NeurIPS'25 | DINOv2 | 98.9 | 98.1 | 99.3 | 89.8 | 90.4 | 89.5 | 91.3 | 93.3 | 90.3 | 93.3 | 93.9 | 93.0 |
| RLCD | ICML'25 | DINOv2 | 99.0 | 98.9 | 99.1 | 91.2 | 91.2 | 91.2 | 92.1 | 96.2 | 90.0 | 94.1 | 95.4 | 93.4 |
| Hyp-SimGCD | CVPR'25 | DINOv2 | 98.9 | 97.7 | 99.5 | 91.5 | 90.0 | 94.6 | 91.9 | 96.2 | 89.8 | 94.1 | 94.6 | 94.6 |
| Hyp-SelEx | CVPR'25 | DINOv2 | 98.6 | 98.1 | 98.9 | 88.6 | 91.5 | 82.8 | 92.3 | 96.4 | 90.2 | 93.2 | 95.3 | 90.6 |
| SimGCD | ICCV'23 | DINOv2 | 98.7 | 96.7 | **99.7** | 88.5 | 89.2 | **87.2** | 89.9 | **95.5** | 87.1 | 92.4 | 93.8 | **91.3** |
| PartCo-SimGCD | **Ours** | DINOv2 | **99.0** | **98.7** | 99.2 | **89.0** | **92.0** | 83.0 | **90.1** | 92.0 | **89.2** | **92.7** | **94.2** | 90.4 |
| SelEx | ECCV'24 | DINOv2 | 98.5 | 98.8 | 98.5 | 87.7 | 90.8 | 81.5 | 90.9 | 96.2 | 88.3 | 92.4 | 95.3 | 89.4 |
| PartCo-SelEx | **Ours** | DINOv2 | **99.2** | **99.4** | **98.9** | **90.0** | **92.8** | **84.3** | **94.5** | **97.8** | **92.8** | **94.6** | **96.7** | **92.0** |
| k-means | - | DINOv3 | 94.1 | 95.1 | 93.6 | 65.5 | 66.4 | 63.5 | 78.3 | 78.3 | 78.3 | 79.3 | 79.9 | 78.5 |
| SimGCD | ICCV'23 | DINOv3 | 98.4 | **98.7** | 98.2 | 84.3 | 88.3 | 74.3 | 92.2 | 96.5 | 90.0 | 91.6 | **94.8** | 87.5 |
| PartCo-SimGCD | **Ours** | DINOv3 | **98.9** | 98.2 | **99.3** | **85.0** | **88.9** | **77.1** | **93.7** | 96.5 | **92.3** | **92.5** | 94.5 | **89.6** |
| SelEx | ECCV'24 | DINOv3 | 98.2 | 99.0 | 97.7 | 87.7 | 88.7 | 85.7 | 93.4 | **96.9** | 91.6 | 93.1 | 94.9 | 91.6 |
| PartCo-SelEx | **Ours** | DINOv3 | **98.3** | **99.1** | **97.9** | **90.2** | **91.7** | **87.0** | **93.8** | 96.6 | **92.4** | **94.1** | **95.8** | **92.4** |

## 4.3 MODEL COMPONENT ANALYSIS

**Effectiveness of 1st vs. 2nd order part-level correspondence labels.** We conduct an ablation study within our PartCo framework to compare 1st, and 2nd order labels, their combination, and a baseline across three fine-grained datasets and one generic dataset (Fig. 7). The results show that 1st order labels consistently achieve the highest accuracy on fine-grained datasets due to their inherently rich part-level information, which remains effective after downsampling. In contrast, 2nd order labels, though detailed, suffer reduced performance on fine-grained datasets because downsampling limits the number of patches per label. However, on generic datasets like ImageNet-100, 2nd order labels outperform 1st order labels by providing more fine-grained information per sample, where 1st order

labels lack sufficient detail. These findings highlight the importance of selecting the appropriate order level based on dataset characteristics, demonstrating PartCo's flexibility in various scenarios.

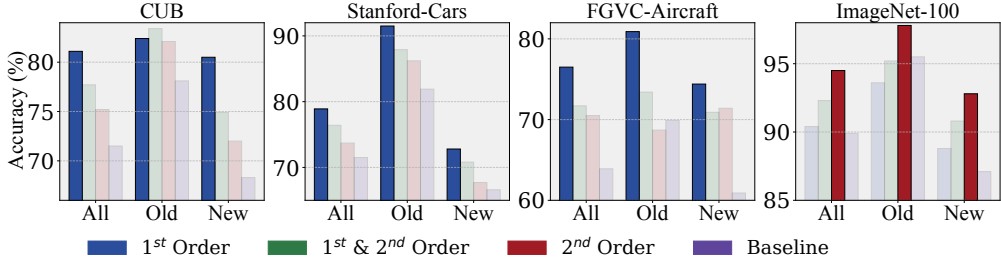

Figure 7: Ablation study investigating the impact of different order levels in part-level correspondence labels. Highest All *ACC* is emphasized, while lower All *ACC* are displayed with reduced opacity.

**Impact of output dimensions on part-level projection.** We conduct an ablation study to evaluate the impact of different output dimension sizes for part-level projections $\psi_p$ within our PartCo framework, as shown in Tab. 3. The results indicate that an output dimension $d'$ of 128 consistently yields the best performance across the CUB and Stanford-Cars datasets, achieving the highest accuracy in the overall category metric.

Table 3: Ablation study on part-level projection output dimensions.

| Dim. | CUB | | | Stanford-Cars | | |
|---|---|---|---|---|---|---|
| | All | Old | New | All | Old | New |
| 64 | 79.3 | 76.7 | 80.6 | 76.9 | 88.3 | 71.4 |
| 128 | **81.1** | 82.4 | 80.5 | **78.9** | 91.5 | 72.8 |
| 256 | 79.8 | 80.7 | 79.4 | 76.4 | 90.5 | 69.7 |
| 512 | 78.2 | 83.1 | 75.7 | 75.6 | 87.5 | 69.9 |

**Impact of balancing factor** $\lambda_b$**.** As detailed in Sec. 3, to effectively leverage our part-level correspondence labels, we design part-level correspondence loss that learns part-level relationship withtin the ViT's patch features. A balancing factor $\lambda_b$ is introduced to regulate the balance between our proposed loss and the baseline's losses. In search of the optimal balancing factor, we investigate a range of weight values, ranging from 0, 0.1, 0.35, 0.7, and 1.0. The results, as shown in Tab. 4, demonstrate that a fixed weight 0.35 is robust and consistently achieve the highest accuracy gains, highlighting PartCo's strong practical generalization.

Table 4: Ablation study on different balancing factor $\lambda_b$ values.

| $\lambda_b$ | CUB | | | Stanford-Cars | | |
|---|---|---|---|---|---|---|
| | All | Old | New | All | Old | New |
| 0 | 71.5 | 78.1 | 68.3 | 71.5 | 81.9 | 66.6 |
| 0.1 | 76.4 | 80.9 | 74.1 | 75.4 | 85.7 | 70.5 |
| 0.35 | **81.1** | 82.4 | 80.5 | **78.9** | 91.5 | 72.8 |
| 0.7 | 79.0 | 82.3 | 77.4 | 77.1 | 88.4 | 71.7 |
| 1.0 | 77.1 | 82.9 | 74.1 | 75.2 | 86.3 | 69.9 |

**Effect of unsupervised part-level correspondence loss.** In addition to the supervised part-level correspondence loss, $\mathcal{L}_{pc}^{sup}$, we conduct an ablation study on PartCo-SimGCD to evaluate the effectiveness of the unsupervised part-level correspondence loss, $\mathcal{L}_{pc}^{unsup}$. As demonstrated in Fig. 8, incorporating $\mathcal{L}_{pc}^{unsup}$ leads to significant improvements in category performance across datasets, highlighting the versatility and robustness of our additional loss when combined with SimGCD.

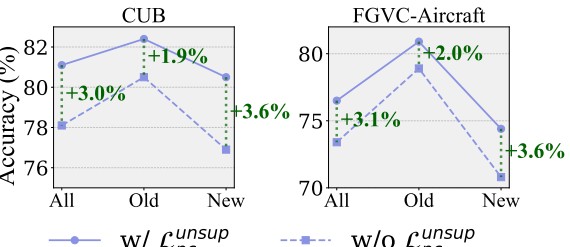

Figure 8: Impact of the unsupervised part-level correspondence loss ($\mathcal{L}_{pc}^{unsup}$).

**Part-level learning boosts category discovery.** We assess how part-level learning improves GCD by adding (i) an implicit part learner (SPTNet (Wang et al., 2024)) and (ii) our explicit framework (PartCo) to the SimGCD baseline (Wen et al., 2023) with DINOv2. Results on CUB and Stanford-Cars datasets are summarized in Tab. 5. As demonstrated, both approaches provide clear benefits. Adding SPTNet to SimGCD improves the overall accuracy. Using PartCo alone brings larger gains (81.1% on CUB; 78.9% on Stanford-Cars). Combining SPTNet with PartCo outperforms other methods, indicating strong complementarity: implicit cues from SPTNet and explicit part constraints from PartCo enhance feature quality and category separation in distinct ways. In summary, our explicit PartCo not only outperforms the baseline and the implicit SPTNet on its own, but also further amplifies the gains of the implicit framework when combined, underscoring that explicit part learning provides complementary supervision that unlocks additional improvements in category discovery.

Table 5: **Implicit vs. explicit part-level learning.** Study on implicit (SPTNet) and explicit (PartCo) part-level learning frameworks on baseline parametric model: SimGCD.

| SimGCD baseline | | CUB | | | Stanford-Cars | | |
|---|---|---|---|---|---|---|---|
| + PartCo (**Ours**) | + SPTNet | All | Old | New | All | Old | New |
| ✗ | ✗ | 71.5 | 78.1 | 68.3 | 71.5 | 81.9 | 66.6 |
| ✗ | ✔ | 76.3 | 79.5 | 74.6 | 72.5 | 82.0 | 67.5 |
| ✔ | ✗ | 81.1 | **82.4** | 80.5 | 78.9 | **91.5** | 72.8 |
| ✔ | ✔ | **82.6** | 82.3 | **81.8** | **80.1** | **92.0** | **73.5** |

## 5 CONCLUSION

In this paper, we introduced PartCo, a learning framework for GCD by integrating explicit part-level visual feature correspondences. Unlike traditional GCD methods that rely solely on semantic labels, PartCo leverages the detailed composition of object features to improve category understanding and discovery. Our experiments on multiple benchmark datasets demonstrate that our framework significantly boosts existing GCD methods. Its seamless integration with current approaches without major modifications highlights its practicality and broad applicability. By focusing on part-level relationships, PartCo not only increases discovery accuracy but also provides deeper insights into the visual structures underlying semantic labels. Overall, PartCo bridges the gap between semantic labels and part-level feature compositions, setting the new SOTA for GCD.

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

# PartCo: Part-Level Correspondence Priors Enhance Category Discovery
## –*Supplementary Material*–

## CONTENTS

## S1 RELATED WORK

**Semi-Supervised Learning (SSL).** SSL aims at learning a classifier using both labeled and unlabeled data (Chapelle et al., 2009; Zhu, 2005; Oliver et al., 2018). Most works in this domain assume that the unlabeled data contains instances from the *same* categories in the labeled data (Oliver et al., 2018). Pseudo-labeling (Rizve et al., 2021), consistency regularization (Laine & Aila, 2017; Tarvainen & Valpola, 2017; Berthelot et al., 2019; Sohn et al., 2020), and non-parametric classification (Assran et al., 2021) are among the popular methods for SSL. Most recent works further remove the assumption on the categories in the unlabeled and labeled set, (Saito et al., 2021; Huang et al., 2021; Yu et al., 2020), yet their focus is still on the performance in the labeled set.

**Feature descriptors with vision foundation models.** Recent ViT models, particularly the DINO variants (Caron et al., 2021; Oquab et al., 2024; Siméoni et al., 2025), have advanced the generation of semantically meaningful and spatially coherent features for dense visual descriptors (Amir et al., 2022). Building on DINOv1 (Caron et al., 2021), DINOv2 (Oquab et al., 2024) scales up the model size and incorporates a larger curated dataset, enhancing generalization in correspondence tasks such as part-level segmentation and zero-shot semantic correspondence (Zhang et al., 2023).

**Part-level learning and semantic correspondence.** Supervised fine-grained pipelines use annotated parts to normalize pose and reduce intraclass variance (Branson et al., 2014), and parts can be transferred between categories with reduced labels (Novotny et al., 2016). To reduce annotation cost, weakly / self-supervised methods learn dense features sensitive to geometry or geometrically stable for semantic matching (Novotny et al., 2017; 2018), while unsupervised approaches induce parts through shape-appearance disentangling or contrastive reconstruction (Lorenz et al., 2019; Choudhury et al., 2021). Cross-category canonicalization learns shared dense geometry without manual inter-category links (Neverova et al., 2021), and open-vocabulary part segmentation scales part reasoning using vision–language supervision and DINO-style descriptors (Sun et al., 2023; Choi et al., 2025). Unlike these lines that focused on part alignment or segmentation, we study *Generalized Category Discovery* (GCD), where a small labeled subset (known classes) coexists with unlabeled data containing both known and novel classes. Prior part-based methods typically assume part/keypoint labels (Novotny et al., 2016) or optimize matching-specific objectives (Novotny et al., 2017); while AnchorNet transfers to unseen classes for matching (Novotny et al., 2017), these works do not discover or cluster unknown categories without labels. In contrast, *PartCo distills part-level observations* into GCD via a tailored correspondence loss, yielding robust, part-aware features under mixed supervision and enabling semantically coherent clusters for unseen categories without part annotations or open-vocabulary text labels, while benefiting from the explicit inductive biases revealed by part-level correspondences.

**Category Discovery.** Novel Class Discovery (NCD) (Han et al., 2019) facilitates knowledge transfer from known to unseen categories through transfer clustering. Since its introduction, various methods are developed to advance NCD (Han et al., 2020; 2021; Jia et al., 2021; Zhao & Han, 2021; Zhong et al., 2021; Fini et al., 2021). Generalized Category Discovery (GCD) (Vaze et al., 2022a) extends NCD by incorporating unlabeled data from both known and unknown classes, presenting additional challenges. Subsequent research on GCD proposes diverse strategies to address these complexities (Cao et al., 2022; Joseph et al., 2022; Pu et al., 2023; Hao et al., 2024; Cendra et al., 2024; Wang et al., 2025c; Liu & Han, 2025; Han et al., 2025). For example, SimGCD (Wen et al., 2023) introduces a parametric classifier with mean entropy regularization, while GPC (Zhao et al., 2023) utilizes Gaussian mixture models to learn robust representations and estimate the number of unknown categories. SPTNet (Wang et al., 2024) employs spatial prompt tuning to enhance focus on specific object parts, improving knowledge transfer in GCD tasks. Recently, FlipClass (Lin et al., 2024) dynamically updates the teacher model to align with the student's attention, ensuring consistency in all classes, SelEx (Rastegar et al., 2024) achieves strong performance in fine-grained datasets through hierarchical semi-supervised $k$-means clustering, NC-GCD (Han et al., 2025) leverages a neural collapse-inspired framework for generalized category discovery, and HypCD (Liu et al., 2025b) achieves SOTA performances by learning representation in hyperbolic space. Additionally, category discovery is explored in various contexts, including multi-modal settings (Jia et al., 2021), continual learning (Zhang et al., 2022; Cendra et al., 2024), federated environments (Pu et al., 2024), and handling domain shifts (Wang et al., 2025c). A concurrent work, APL (Dai et al., 2025) , also uses part-level cues for GCD by learning part queries guided by DINO priors and replacing the backbone's classification features with an aggregated part representation. In contrast, our method,

PartCo, introduces explicit part-level correspondence labels to improve category discovery without modifying the original design of the baseline model, highlighting our simple and flexible framework. This approach not only improves the accuracy and robustness of discovering novel categories but also provides a more nuanced understanding of category relationships through finer-grained semantic structures, offering a novel framework for category discovery.

## S2 ADDITIONAL EXPERIMENTAL DETAILS

### S2.1 BENCHMARK DATASETS

For each benchmark dataset, we follow the data splitting approach outlined in Vaze et al. (2022a). In this approach, 50% of the classes are designated as 'Old', except for CIFAR-100, which selects 80% of the classes. Subsequently, 50% of the images from the known classes form the labeled dataset $\mathbf{D}_l$, while the remaining images are assigned to the unlabeled dataset $\mathbf{D}_u$. The statistics of all datasets used in this study are presented in Tab. A.

### S2.2 TIME ANALYSIS FOR PART-LEVEL CORRESPONDENCE LABELS CONSTRUCTION

In Sec. 3.1 of the main paper, we describe the methodology for generating part-level correspondence labels. The overall time required to construct these labels for each dataset is detailed in the last column of Tab. A. As illustrated, our label construction process is efficient, with execution times ranging from approximately 5 to 180 minutes depending on the dataset's size and complexity. Importantly, this time cost is negligible compared to typical model training durations, and it is incurred solely during the training phase. Consequently, the label construction does not introduce any additional latency during the inference phase. While it is feasible to integrate the label construction directly into the training pipeline, we have opted to separate these steps to simplify the implementation and facilitate easier experimentation.

Table A: **Dataset statistics.** We specify the number of classes in the labeled and unlabeled sets as $M = |\mathbf{Y}_l|$ and $K = |\mathbf{Y}_l \cup \mathbf{Y}_u|$, respectively, along with the image counts $|\mathbf{D}_l|$ and $|\mathbf{D}_u|$. In the last column, we also provide the time taken (minutes) to construct PartCo's labels.

| Dataset | $|\mathbf{D}_l|$ | $M$ | $|\mathbf{D}_u|$ | $K$ | PartCo label time (min) |
|---|---|---|---|---|---|
| CUB (Wah et al., 2011) | 1.5K | 100 | 4.5K | 200 | 6 m |
| Stanford-Cars (Krause et al., 2013) | 2.0K | 98 | 6.1K | 196 | 5 m |
| FGVC-Aircraft (Maji et al., 2013) | 1.7K | 50 | 5.0K | 100 | 7 min |
| CIFAR10 (Krizhevsky & Hinton, 2009) | 12.5K | 5 | 37.5K | 10 | 30 m |
| CIFAR100 (Krizhevsky & Hinton, 2009) | 20.0K | 80 | 30.0K | 100 | 30 m |
| ImageNet-100 (Deng et al., 2009) | 31.9K | 50 | 95.3K | 100 | 180 m |
| Oxford-Pet (Parkhi et al., 2012) | 0.9K | 19 | 2.7K | 37 | 5 m |
| Herbarium19 (Tan et al., 2019) | 8.9K | 341 | 25.4K | 683 | 118 m |

### S2.3 COMPUTATIONAL OVERHEAD ANALYSIS

We report concrete training and compute statistics in Tab. B. Compared to SimGCD, PartCo-SimGCD incurs only modest overhead on CUB / Stanford Cars / CIFAR100: training time increases by just 5%, 2.5%, and 4.2% respectively, peak memory (with 128 batch size) just rises from 8.6 GB to 12.0 GB, and GFLOPs grow by just only 15%. This extra cost mainly comes from computing patch-level correspondences and the additional patch projection head. Importantly, inference remains unchanged: at test time, PartCo uses the same backbone features and clustering pipeline as the underlying baseline methods. By introducing this small computation overhead over the baseline, PartCo significantly improve current baselines across benchmarks and pretrained backbones.

Table B: **Computational overhead results.** Computational overhead of PartCo-SimGCD relative to SimGCD: modest increases in training time, peak memory, and GFLOPs; inference pipeline remains unchanged.

| Method | Training time | | | Peak training memory usage | GFLOPs |
|---|---|---|---|---|---|
| | CUB | Stanford Cars | CIFAR100 | | |
| SimGCD (baseline) | 3h 22m | 4h 45m | 30h 7m | 8.6 GB | 54.1 |
| PartCo-SimGCD (**Ours**) | 3h 32m (+5%) | 4h 52m (+2.5%) | 31h 22m (+4.2%) | 12.0 GB | 62.6 (+15%) |

# S3 ADDITIONAL QUANTITATIVE RESULTS

## S3.1 EXPERIMENTS ON ADDITIONAL DATASETS

To further assess the effectiveness of our PartCo framework, we conducted evaluations on two additional fine-grained datasets: Oxford-Pet (Parkhi et al., 2012) and Herbarium19 (Tan et al., 2019). The Oxford-Pet dataset is particularly challenging due to its diverse assortment of cat and dog species combined with limited data availability. In contrast, Herbarium19 is a botanical research dataset that includes a wide variety of plant types, characterized by its long-tailed distribution and detailed categorization. The details of these two datasets are shown in Tab. A.

As shown in Tab. C, our PartCo-enhanced models consistently outperform the baseline method across all categories. Specifically, PartCo-SimGCD achieves an impressive accuracy of 95.2% on the 'All' category of the Oxford-Pet dataset, significantly surpassing the SimGCD baseline's 86.2%. Similarly, on the Herbarium19 dataset, PartCo-SimGCD attains an accuracy of 55.5%, outperforming the baseline's 48.6%. Overall, these results demonstrate that our PartCo framework effectively enhances existing baseline models, even when applied to more challenging fine-grained and long-tailed datasets.

Table C: **Enhancement of baseline GCD methods with PartCo framework.** Performance on the Oxford-Pet (Parkhi et al., 2012) and Herbarium19 (Tan et al., 2019) datasets using DINOv2. Results are reported in *ACC* across the 'All', 'Old' and 'New' categories. † denotes results implemented by us.

| Method | Oxford-Pet | | | Herbarium19 | | |
|---|---|---|---|---|---|---|
| | All | Old | New | All | Old | New |
| SimGCD | 86.2 | 85.4 | 86.6 | 48.6 | 64.8 | 39.9 |
| PartCo-SimGCD (**Ours**) | **95.2** | **92.7** | **96.6** | **55.5** | **68.0** | **48.7** |
| FlipClass† | 91.4 | 88.0 | 93.2 | 55.8 | 68.1 | 49.2 |
| PartCo-FlipClass (**Ours**) | **95.3** | **93.7** | **96.1** | **57.4** | **69.0** | **51.2** |
| SelEx† | 91.5 | 96.7 | 88.6 | 43.1 | 54.1 | 37.2 |
| PartCo-SelEx (**Ours**) | **92.7** | **96.9** | **90.4** | **45.9** | **57.3** | **39.7** |

## S3.2 COMPARISON OF DIFFERENT APPROACHES FOR PART-LEVEL LABEL CONSTRUCTION

In this work, we use PCA + DINOv2 (Oquab et al., 2024) features for the following reasons: **(1) Generalization.** PCA with DINOv2 offers excellent off-the-shelf generalization for generating part-level correspondences without the need for additional tuning or adaptation. According to Zhang et al. (2023), DINOv2 outperforms other foundation models like DINO and Stable Diffusion (SD) models, and is only slightly less effective than combining DINOv2 + SD. **(2) Efficiency.** Incorporating SD-based features, or a combination of SD and DINO-based models, significantly increases computational costs and memory usage (Zhang et al., 2023). This makes the part-label construction process inefficient. By using DINOv2 solely, our approach remains both computationally and model-efficient.

Moreover, we construct labels by augmenting DINOv2 features with SD features to compute PCA, following Zhang et al. (2023). Because SD requires an inference denoising step, this procedure is computationally heavy: on CUB it takes around 108 minutes, whereas our PCA + DINOv2 pipeline takes around 6 minutes (Tab. A, supp. material), making SD-DINOv2 impractical for larger datasets (e.g., ImageNet-100). We further compared performance using SD-DINOv2-based labels against our original labels. As shown in Tab. D, the 'All' *ACC* differences are marginal (about 0.2–0.6 percentage points) while our PCA + DINOv2 label construction is substantially faster and more efficient.

## S3.3 STUDY ON UNKNOWN CATEGORY ESTIMATION METHODS

In the real-world category discovery task, the exact number of novel categories is often unknown, posing a significant challenge for model training and evaluation. Existing literature (Vaze et al., 2022a; Hao et al., 2024; Zhao et al., 2023), has explored methods to estimate the number of unknown categories ($K$). Building upon these studies, we conduct a comprehensive analysis to assess the effectiveness of different $K$-estimation methods when integrated with a stronger foundation model, specifically DINOv2 (Oquab et al., 2024).

Table D: **Influence of different part-level correspondence labels construction techniques.** Performance on the CUB and Stanford-Cars datasets with different part-level correspondence labels generated by Our method vs. SD-DINOv2 (Zhang et al., 2023). Results are reported in *ACC* across the 'All', 'Old' and 'New' categories.

| PartCo-SimGCD | CUB | | | Stanford-Cars | | |
|---|---|---|---|---|---|---|
| | All | Old | New | All | Old | New |
| w/ DINOv2 (**Ours**) | 81.1 | 82.4 | 80.5 | 78.9 | 91.5 | 72.8 |
| w/ SD-DINOv2 | 80.9 | 79.7 | 81.6 | 78.3 | 90.0 | 72.9 |

Our experimental setup involves training three distinct GCD methods, *i.e.*, GCD (Vaze et al., 2022a), SelEx (Rastegar et al., 2024), and our proposed PartCo-SelEx (Ours) using DINOv2 pretrained weights. These trained models serve as feature extractors for the subsequent $K$-estimation process. We employ two off-the-shelf $K$-estimation techniques: GCD (Vaze et al., 2022a) and CiPR (Hao et al., 2024) $K$-est methods. The performance of these integrated approaches is evaluated on two fine-grained datasets: CUB (Wah et al., 2011) and Stanford-Cars (Krause et al., 2013).

Table E: **Unknown category estimation.** Category estimation results of various $K$-estimation methods on different GCD methods using DINOv2.

| Method | CUB | | Stanford-Cars | |
|---|---|---|---|---|
| | GCD $K$-est | CiPR $K$-est | GCD $K$-est | CiPR $K$-est |
| GCD | 188 | 178 | 242 | 169 |
| SelEx | 219 | 191 | 229 | 194 |
| PartCo-SelEx (**Ours**) | 210 | 192 | 185 | 195 |
| *Ground-truth K* | 200 | | 196 | |

Tab. E shows the estimation results of different $K$-estimation methods when applied to the GCD, SelEx, and PartCo-SelEx methods. The ground-truth number of categories is 200 for the CUB dataset and 196 for the Stanford-Cars dataset. We observe that the GCD $K$-estimation method, when paired with the GCD's weights, significantly underestimates $K = 188$ for CUB dataset and overestimates $K = 242$ for Stanford-Cars. In contrast, the CiPR $K$-estimation method offers improved estimations, though still not perfectly aligned with the ground truth ($K = 178$ for CUB and $K = 169$ for Stanford-Cars).

When integrating SelEx with the $K$-estimation methods, the performance improves, with CiPR providing more accurate estimates ($K = 191$ for CUB and $K = 194$ for Stanford-Cars) compared to GCD's native method ($K = 219$ for CUB and $K = 229$ for Stanford-Cars). On the other hand, our proposed PartCo-SelEx framework demonstrates the most accurate $K$-estimation across both datasets, achieving estimates of 210 for CUB and 185 for Stanford-Cars with the GCD $K$-estimation method, and $K = 192$ for CUB and $K = 195$ for Stanford-Cars with the CiPR method. These results indicate that PartCo-SelEx consistently provides $K$-estimates that are closer to the ground truth, particularly when using the CiPR estimation method.

Table F: **GCD performance with estimated K.** GCD results on DINOv2 with the estimated number of categories.

| Method | CUB | | | Stanford-Cars | | |
|---|---|---|---|---|---|---|
| | All | Old | New | All | Old | New |
| GCD | 68.9 | 77.0 | 65.0 | 62.2 | 72.5 | 57.2 |
| SimGCD | 70.4 | 78.1 | 66.7 | 69.7 | 84.8 | 62.3 |
| PartCo-SimGCD (**Ours**) | 78.8 | **80.3** | 78.0 | 73.9 | 84.9 | 68.6 |
| SelEx | 86.1 | 77.8 | 90.3 | 78.3 | **89.2** | 73.0 |
| PartCo-SelEx (**Ours**) | **87.6** | 79.3 | **91.7** | **80.6** | 88.7 | **76.8** |

To further evaluate the robustness of our method under challenging $K$-estimation scenarios, we conduct experiments using the worst estimation method results: $K = 188$ for CUB and $K = 242$ for Stanford-Cars, as shown in Tab. F. Despite these inaccurate $K$-estimates, our PartCo-SelEx framework maintains superior performance compared to baseline models. Specifically, on the CUB dataset, PartCo-SelEx achieves an accuracy of 87.6% on the 'All' category and 91.7% on the 'New' category, outperforming the SelEx baseline which achieves 86.1% and 90.3% respectively. Similarly, on the Stanford-Cars dataset, PartCo-SelEx attains the highest accuracies of 80.6% for 'All' and

76.8% for 'New' categories, surpassing the SelEx baseline's 78.3% and 73.0% respectively. These results underscore the robustness of PartCo-SelEx in handling erroneous $K$-estimates, ensuring consistent and reliable performance even when the estimated number of categories deviates from the ground truth.

## S3.4 EXPERIMENT ON CLIP BACKBONE

To assess PartCo's generalization beyond DINO variants, we run additional experiments using only CLIP (Radford et al., 2021) ViT-B/16 vision encoder (see Tab. G). Even with the image encoder alone, PartCo consistently and substantially improves the SimGCD baseline across multiple datasets, yielding strong overall performance (with $\sim 5\%$ accuracy gains). Note that competing methods (Zheng et al., 2024; Wang et al., 2025b; Ouldnoughi et al., 2023; Yang et al., 2025) use both CLIP's image and text encoders. This suggests that our framework does not hinge on a particular pretraining strategy (*e.g.*, DINO) but instead leverage a more general property of modern vision transformers: their ability to provide reasonably structured patch-level features. As with any method that builds on a pretrained backbone, PartCo of course will inherit some biases of the underlying foundation model. However, the empirical results with both DINOv2/v3 and CLIP indicate that PartCo is robust across different pretrained backbone, and that its benefits are not confined to a single pretrained family.

Table G: Comparison of GCD methods on the SSB benchmark datasets using CLIP backbone. Results are reported in *ACC* across the 'All', 'Old' and 'New' categories.

| Method | Encoder | Backbone | CUB | | | Stanford-Cars | | | FGVC-Aircraft | | | Average | | |
|---|---|---|---|---|---|---|---|---|---|---|---|---|---|---|
| | | | All | Old | New | All | Old | New | All | Old | New | All | Old | New |
| GCD | Image | CLIP | 57.6 | 65.2 | 53.8 | 65.1 | 75.9 | 59.8 | 45.3 | 44.4 | 45.8 | 56.0 | 61.8 | 53.1 |
| TextGCD | Image & Text | CLIP | 76.6 | 80.6 | 74.7 | 86.9 | 87.4 | 86.7 | - | - | - | - | - | - |
| GET | Image & Text | CLIP | 77.0 | 78.1 | 76.4 | 78.5 | 86.8 | 74.5 | 58.9 | 59.6 | 58.5 | 71.5 | 74.8 | 69.8 |
| CLIP-GCD | Image & Text | CLIP | 62.8 | 77.1 | 55.7 | 70.6 | 88.2 | 62.2 | 50.0 | 56.6 | 46.5 | 61.1 | 74.0 | 54.8 |
| CPT | Image & Text | CLIP | 70.1 | 73.5 | 68.4 | 74.2 | 84.3 | 69.3 | - | - | - | - | - | - |
| SimGCD | Image | CLIP | 71.7 | 76.5 | 69.4 | 70.0 | 83.4 | 63.5 | 54.3 | 58.4 | 52.2 | 65.3 | 72.8 | 61.7 |
| PartCo-SimGCD (**Ours**) | Image | CLIP | **73.4** | **80.2** | **70.0** | **76.5** | **89.6** | **70.2** | **61.6** | **62.9** | **60.9** | **70.5** | **77.6** | **67.0** |

## S3.5 FURTHER ANALYSIS ON PART-LEVEL CORRESPONDENCE LABELS

**Comparison between PartCo's labels and annotated part labels.** To verify the effectiveness and quality of our part-level correspondence labels, we compare our constructed CUB's part correspondence labels (before downsample) against the manually annotated parts provided by CUB dataset and report the results in Tab. H. For each visible ground-truth part, we check whether the assigned part labels matches the corresponding CUB part through hungarian matching algorithm. The results show that our 1st and 2nd order part-level correspondence labels achieve overall accuracies of 80.8% and 77.5%, respectively, with especially strong alignment on key semantics parts such as *back, beak, belly, leg, wing, and tail*. Moreover, the significant performance gains reported in the main paper (Tab. 1 and Tab. 2) suggest that the part-level correspondence labels are of sufficiently high quality to provide a reliable supervisory signal for our part-level correspondence learning framework.

Table H: Comparison with CUB ground-truth part labels.

| CUB's part index | No. visible parts | CUB's part-level correspondence labels | |
|---|---|---|---|
| | | 1st part labels *ACC* | 2nd part labels *ACC* |
| Part 1: back | 9064 | 96.2 | 97.5 |
| Part 2: beak | 11745 | 97.9 | 91.0 |
| Part 3: belly | 10347 | 76.2 | 89.2 |
| Part 5: crown | 11580 | 86.4 | 89.8 |
| Part 6: forehead | 11603 | 50.2 | 43.1 |
| Part 7: leg (left/right) | 17008 | 85.2 | 75.7 |
| Part 8: wing (left/right) | 13255 | 67.6 | 69.7 |
| Part 10: tail | 10961 | 89.5 | 70.9 |
| *Overall* | 95563 | 80.8 | 77.5 |

**Why further expanding PartCo's order levels hurts category discovery.** In the main paper, we show that first-order part labels and second-order part labels yield the best gains on fine-grained and coarse-grained datasets respectively (see Fig. 7). In this section, we ask whether pushing to even higher orders continues to help. To examine this, we extend PartCo to 3rd order part-level

correspondences by mirroring our second-order construction: we apply an additional PCA on the fine-grained features within each second-order cluster to obtain finer partitions. As visualized for CUB in Fig. A, this produces over-fragmentation: the same semantic part (*e.g.*., a bird's head or wing) is split across several distinct labels, yielding redundant, noisy correspondences that are less semantically meaningful and harder to transfer.

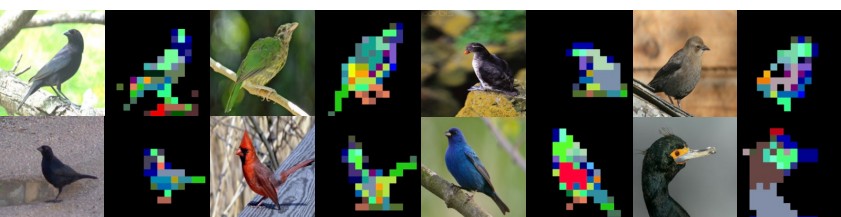

Figure A: Additional visualization of our third-order part-level correspondence labels.

We further quantitatively examine this in Tab. I, by utilizing $3^{rd}$ order labels in our framework for CUB and ImageNet100 datasets; performance consistently drops. With a fixed token budget in the transformer (*e.g.*, 196 patch tokens), the over-clustered supervision spreads attention too thin, impeding effective part-level correspondence learning compared to our current design. Taken together, these results suggest a practical principle: $1^{st}$ order and $2^{nd}$ order correspondences strike the right balance, capturing part identity for both fine-grained and generic settings, whereas higher orders over-cluster and leads to ineffective part corresponding learning.

Table I: **Ablation of part-level order granularity.** Extending PartCo to third-order labels consistently hurts accuracy across datasets; first-order and second-order labels provide the best improvement gains for finegrained and generic datasets respectively.

| Method | CUB | | | ImageNet100 | | |
|---|---|---|---|---|---|---|
| | All | Old | New | All | Old | New |
| baseline | 71.5 | 78.1 | 68.3 | 89.9 | 95.5 | 87.1 |
| +$1^{st}$ order labels | **81.1** | **82.4** | **80.5** | 90.4 | 93.6 | 88.8 |
| +$2^{nd}$ order labels | 75.2 | 82.1 | 72.0 | **94.6** | **96.7** | **92.0** |
| +$3^{rd}$ order labels | 72.7 | 81.3 | 68.4 | 91.8 | 94.6 | 90.4 |

## S3.6 EVALUATING BACKGROUND SUPPRESSION VS. EXPLICIT PART CORRESPONDENCE

To verify whether explicit part modeling is necessary, we add ablation where we simply mask background regions on both labeled and unlabeled images using our background mask obtained during our part-level correspondence label construction and train the baselines using foreground-only features, without any explicit part correspondence learning. As shown in Tab. J, for SimGCD baseline tested on CUB and Stanford Cars, the foreground-only variant yields only modest gains over the original baseline (*e.g.*, CUB: 71.5 → 73.7 on "All", and Stanford Cars: 71.5 → 73.0 on "All"), indicating that removing background alone provides limited benefits. In contrast, introducing explicit part-level correspondence prior via PartCo framework leads to substantially larger improvements on the same backbones and dataset (see Tab. 1 and Tab. 2), showing that the advantage does not come merely from suppressing background noise but from explicitly aligning semantically meaningful object parts across images.

Table J: Foreground-only ablation vs. PartCo explicit part correspondence learning.

| Method | CUB | | | Stanford Cars | | |
|---|---|---|---|---|---|---|
| | All | Old | New | All | Old | New |
| SimGCD (baseline) | 71.5 | 78.1 | 68.3 | 71.5 | 81.9 | 66.6 |
| SimGCD (foreground only) | 73.7 | 75.4 | 72.8 | 73.0 | 85.6 | 66.9 |
| PartCo-SimGCD (**Ours**) | **81.1** | **82.4** | **80.5** | **78.9** | **91.5** | **72.8** |

## S4 QUALITATIVE RESULTS

### S4.1 ADDITIONAL VISUALIZATION OF PART-LEVEL CORRESPONDENCE LABELS

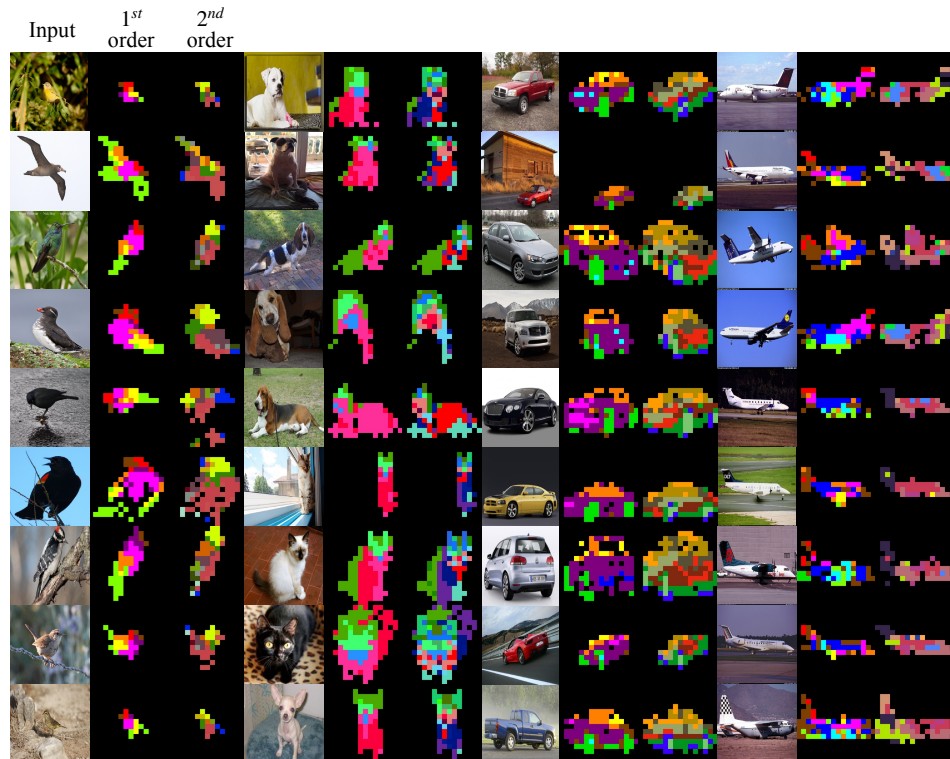

Figure B: **Additional visualization of our part-level correspondence labels.** For every input image, both first and second-order labels are constructed.

### S4.2 QUALITATIVE ANALYSIS OF PARTCO ATTENTION MAPS

We provide a qualitative analysis of the attention maps generated by SimGCD (Wen et al., 2023) and SelEx (Rastegar et al., 2024) when integrated with the PartCo framework (Ours), applied to the CUB (Wah et al., 2011) and Stanford-Cars (Krause et al., 2013) datasets, as shown in Fig. C & D. These attention maps originate from the final block of DINOv2 ViT backbone, utilizing a resolution of $16 \times 16$. Following the methodology outlined in Caron et al. (2021), we calculate the mean value across all attention heads and upsample the resulting maps to the original image resolution for visualization purposes. The analysis shows that all evaluated methods focus their attention on specific parts of the objects, effectively highlighting regions crucial for distinguishing between fine-grained categories. Notably, when combined with SimGCD, the PartCo framework (PartCo-SimGCD) tends to concentrate on particular parts of the object, such as the wings of a bird in the CUB dataset or the wheels of a car in the Stanford-Cars dataset. This targeted focus underscores PartCo-SimGCD's ability to hone in on key discriminative features essential for accurate category differentiation. In contrast, integrating PartCo with SelEx (PartCo-SelEx) results in attention maps that cover a broader range of object parts, capturing multiple fine-grained details simultaneously. For example, PartCo-SelEx not only highlights the wings but also the body and head of the bird in the CUB dataset, and the wheels, doors, and headlights of the car in the Stanford-Cars dataset.

These observations indicate that while both PartCo-SimGCD and PartCo-SelEx effectively utilize part-level information to enhance attention mechanisms, PartCo-SelEx exhibits a more comprehensive focus on multiple object parts. This broader attention coverage can potentially lead to a more nuanced understanding of fine-grained categories, thereby improving the model's ability to generalize across diverse and complex datasets. Overall, the qualitative analysis demonstrates the robustness and effectiveness of the PartCo framework in refining attention maps, highlighting its superior capability to capture meaningful visual regions.

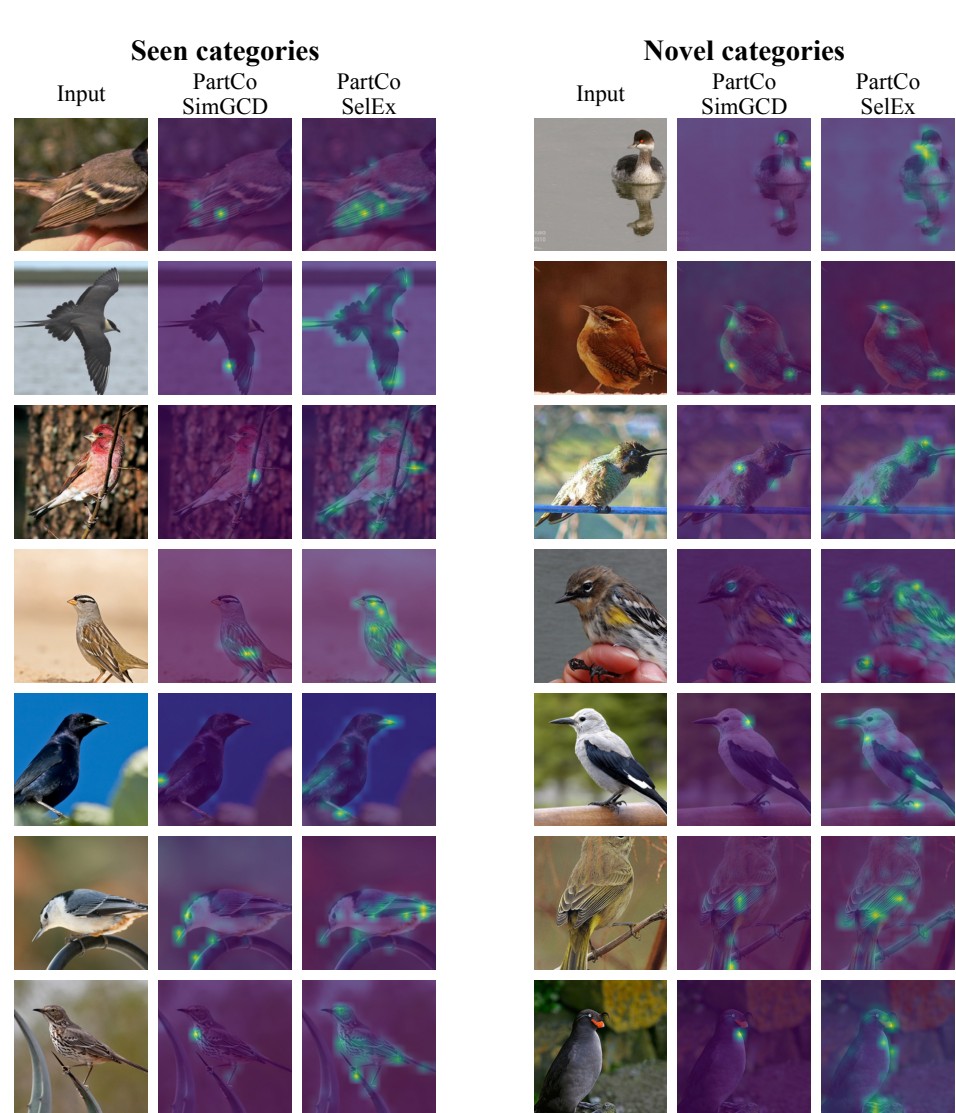

Figure C: **Attention maps on CUB dataset.** Visualization of attention maps generated by the PartCo framework integrated with SimGCD and SelEx for the CUB dataset. The PartCo-SimGCD model highlights specific regions such as the wings and head of the bird, indicating focused attention on key discriminative features. In contrast, the PartCo-SelEx model displays a broader attention distribution, encompassing multiple parts including the wings, body, and tail.

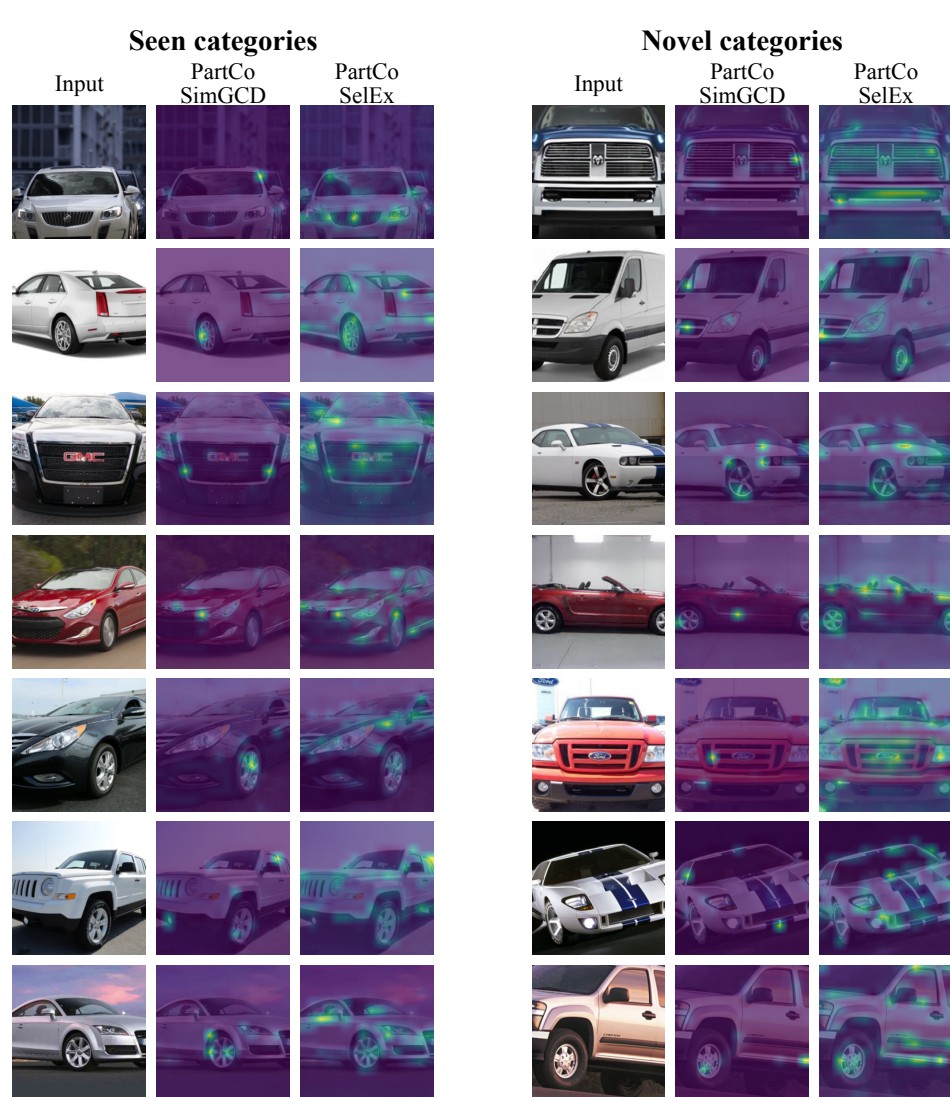

Figure D: **Attention maps on Stanford-Cars dataset.** Visualization of attention maps generated by the PartCo framework integrated with SimGCD and SelEx for the Stanford-Cars dataset. The PartCo-SimGCD model concentrates on distinct parts like the wheels and headlights of the car, demonstrating targeted attention on essential distinguishing features. Meanwhile, the PartCo-SelEx model exhibits a wider area of focus, covering various components such as the wheels, doors, and overall body structure.

## S5   SUCCESS & FAILURE CASE ANALYSIS

Tab. K reports the per-class accuracy difference between PartCo and the Baseline on CUB, highlighting the 10 classes (see Fig. E) with the largest gains ("Success") and the 10 classes (see Fig. F) with the largest drops ("Failure"). We observe that the dominant pattern in the success cases is that PartCo helps on fine-grained, part-dependent species, while the failures largely correspond to classes where global shape or coarse appearance is more informative than local details.

**Success cases (part-dependent classes).** The classes where PartCo substantially outperforms the Baseline (*e.g.*, *Magnolia Warbler, Marsh Wren, Black-throated Blue Warbler, Seaside Sparrow, Barn Swallow*) are mostly small passerines with very similar global silhouettes but distinctive, localized plumage. These species are hard to distinguish using only a global descriptor because they all look like "small round songbirds" in terms of overall shape. Instead, they are identified by specific local cues: for instance, *Magnolia Warbler* (ID: 169) has a characteristic tail and yellow underparts, while Black-throated *Blue Warbler* (ID: 160) is defined by a dark throat patch and contrasting flanks. By explicitly learning and matching part-level features, PartCo can anchor its predictions on these discriminative regions, turning many near-0% Baseline accuracies into high per-class accuracy. Notably, several of these gains occur on unseen classes, indicating that part-level cues provide a transferable signal that generalizes beyond the seen taxonomy.

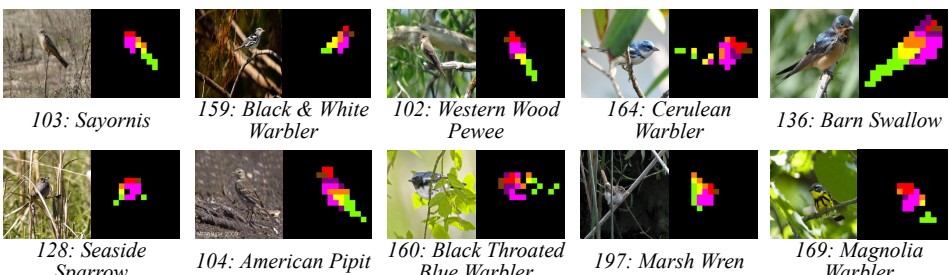

*103: Sayornis*    *159: Black & White Warbler*    *102: Western Wood Pewee*    *164: Cerulean Warbler*    *136: Barn Swallow*

*128: Seaside Sparrow*    *104: American Pipit*    *160: Black Throated Blue Warbler*    *197: Marsh Wren*    *169: Magnolia Warbler*

Figure E: **Visualization of the top 10 success cases on the CUB dataset.** We display sample images for each class alongside their corresponding part-level correspondence labels.

Table K: **Per-class success and failure analysis.** We report the 10 classes with the largest accuracy gains (*Success*, top) and the 10 classes with the largest drops (*Failure*, bottom) when replacing the Baseline with PartCo, along with their seen/unseen type labels. Δ denotes the *ACC* difference.

| ID | Class Name | Type category | ACC Baseline | +PartCo | Δ |
|----|-----------|---------------|---------|---------|---|
| **Success Cases (top 10)** | | | | | |
| 169 | Magnolia Warbler | Seen | 0.0 | 100.0 | +100.0 |
| 197 | Marsh Wren | Seen | 3.3 | 100.0 | +96.7 |
| 160 | Black-throated Blue Warbler | Unseen | 0.0 | 76.7 | +76.7 |
| 164 | Cerulean Warbler | Seen | 0.0 | 75.0 | +75.0 |
| 128 | Seaside Sparrow | Unseen | 0.0 | 73.3 | +73.3 |
| 136 | Barn Swallow | Seen | 21.3 | 93.3 | +72.0 |
| 104 | American Pipit | Unseen | 6.7 | 70.0 | +63.3 |
| 102 | Western Wood Pewee | Unseen | 40.0 | 90.0 | +50.0 |
| 159 | Black & White Warbler | Seen | 46.7 | 96.7 | +50.0 |
| 103 | Sayornis | Unseen | 53.3 | 100.0 | +46.7 |
| **Failure Cases (top 10)** | | | | | |
| 129 | Song Sparrow | Unseen | 76.7 | 41.7 | -35.0 |
| 89 | Hooded Merganser | seen | 63.3 | 33.3 | -30.0 |
| 162 | Canada Warbler | Seen | 83.3 | 58.3 | -25.0 |
| 134 | Cape Glossy Starling | Seen | 100.0 | 75.0 | -25.0 |
| 110 | Geococcyx | Unseen | 73.3 | 50.0 | -23.3 |
| 180 | Wilson's Warbler | Seen | 96.7 | 75.0 | -21.7 |
| 120 | Fox Sparrow | Seen | 80.0 | 60.0 | -20.0 |
| 130 | Tree Sparrow | Seen | 96.7 | 76.7 | -20.0 |
| 32 | Mangrove Cuckoo | Seen | 94.7 | 75.5 | -19.2 |
| 184 | Louisiana Waterthrush | Unseen | 83.3 | 65.0 | -18.3 |

**Failure cases (global/shape-biased or ambiguous classes).** The main failures (*e.g.*, *Song Sparrow, Hooded Merganser, Mangrove Cuckoo, Wilson Warbler*) tend to be classes where (i) global structure is more distinctive than any single local patch, or (ii) local patches are visually ambiguous across species. For instance, *Mangrove Cuckoo* (ID: 32) has a very characteristic elongated body and tail that are easily captured by a global representation, while its local textures (brown and grey feathers) are relatively generic. Similarly, *Wilson Warbler* (ID: 180) is almost uniformly yellow with a small dark cap; zoomed-in patches mostly look like "yellow feathers", which are hard to separate from other yellow warblers. In such cases, forcing the model to over-emphasize parts can down-weight useful global shape cues and make the decision boundary noisier. For *Hooded Merganser* (ID: 89), qualitative inspection of CUB samples, as shown in Fig. G, suggests substantial appearance variation between juveniles and adults, which further increases intra-class variability at the part level and is not explicitly modeled in our current design.

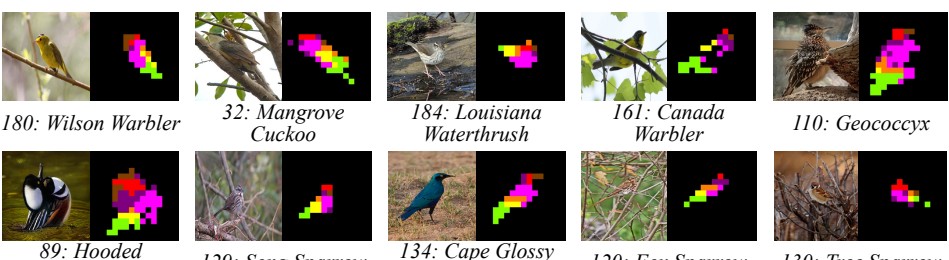

*180: Wilson Warbler*    *32: Mangrove Cuckoo*    *184: Louisiana Waterthrush*    *161: Canada Warbler*    *110: Geococcyx*

*89: Hooded Merganser*    *129: Song Sparrow*    *134: Cape Glossy Starling*    *120: Fox Sparrow*    *130: Tree Sparrow*

Figure F: **Visualization of the top 10 failure cases on the CUB dataset.** We display sample images for each class alongside their corresponding part-level correspondence labels.

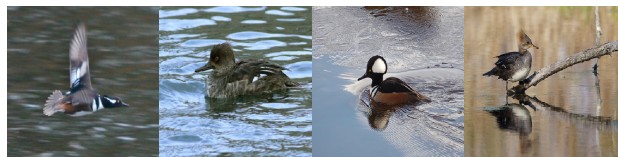

Figure G: **Example *Hooded Merganser* (ID 89) images from CUB,** showing large intra-class variation (juvenile vs. adult, biological development). This variation makes part-level cues ambiguous and contributes to PartCo's reduced accuracy on this class compared to the Baseline.

Overall, however, the magnitude of the largest drops is clearly smaller than the strongest gains (e.g., maximum decrease of around 35 points vs. gains up to 100 points), and several of the biggest improvements occur on unseen classes. This supports our main claim that part-level cues are an effective vehicle for transferring fine-grained discriminative knowledge from seen to unseen categories, while suggesting that future work could mitigate the residual failures by better handling intra-class appearance factors such as age and biological species-specific development.

## S6 LIMITATIONS

The PartCo framework currently relies on foundation models that provide patch token representations, which are characteristic of recent transformer-based architectures. This dependence makes PartCo incompatible with models that lack patch tokens, such as certain convolutional neural networks and older architectures, thereby limiting its generalizability. Future research should focus on developing methods to integrate part-level or localized information into PartCo, allowing it to be effectively applied across a broader range of foundation models.

## S7 BROADER IMPACTS

The development of the Part-Level Correspondence Prior (PartCo) framework represents a significant advancement in category discovery, especially in recognizing fine-grained categories and enabling more robust intelligent systems in real-world scenarios. Fine-grained category recognition is essential in fields such as biodiversity conservation, where precise species identification supports ecological monitoring, and in healthcare, where detailed analysis of medical images could potentially lead to improved disease diagnosis and personalized treatment plans. However, the deployment of PartCo also presents potential ethical and societal challenges. Enhanced image recognition capabilities could be misused in surveillance systems, raising significant privacy concerns. There is a risk that biased training data may lead to unfair or discriminatory outcomes. To mitigate these risks, it is essential to implement robust data governance frameworks that ensure diversity and representativeness in training datasets, thereby minimizing biases. Privacy-preserving techniques and strict regulatory compliance should be prioritized to protect individual rights and prevent misuse in categorization tasks.

