# OpenReview forum: "PartCo: Part-Level Correspondence Priors Enhance Category Discovery"
_ICLR.cc/2026/Conference — Submitted to ICLR 2026_

### Official Review · Reviewer_rMXF · 2025-10-24

**Soundness:** 3
**Presentation:** 3
**Contribution:** 3
**Rating:** 4
**Confidence:** 3

**Summary:**

This paper tackles the task of generalized category discovery - given a labelled training set up some categories learn a model to automatically cluster an unlabelled test set which contains both seen and unseen categories. Particularly this paper focuses on part-level correspondence aiming to capture relationships between categories in terms of shared components/parts to aid generalized category discovery. This is achieved by performing PCA on DINO features to obtain fine-grained features and then clustering these features into parts. The resulting part-level features can then be integrated into existing GCD methods.

**Strengths:**

- The paper is generally well-presented, with visuals that effectively aid understanding of the proposed approach.
***

- The idea of leveraging object parts for GCD is interesting and particularly relevant for fine-grained scenarios where category discovery is more realistic and challenging.
***

- The method is well-motivated, making good use of the richer feature representations available in DINO beyond the CLS token.
***

- The evaluation includes both generic and fine-grained datasets, demonstrating reasonable generality of the approach.
***

- The proposed framework is flexible and can be combined with different existing GCD methods, increasing its potential applicability.
***

- The use of DINOv3 features is appreciated as it helps keep the experiments aligned with the current state of the field.
***

- The comparison between implicit and explicit part-level learning in Table 4 is a nice inclusion and provides some insight into the benefits of the explicit formulation.

**Weaknesses:**

- An existing work in GCD also uses part-level correspondence [A]. The difference to this work in terms of approach and performance needs to be made clear
***

- It remains unclear whether explicit part modelling is necessary. Could similar benefits be achieved simply by removing background features? Including a “single-part” or “foreground-only” variant in the ablation (e.g., Figure 7) would help clarify this.
***

- It would be useful to better understand how the method performs on more challenging or long-tailed datasets such as Herbarium. While Herbarium and Oxford Pets are tested in the supplementary material, the comparison is only made against SimGCD. This feels incomplete, especially since other works such as SelEX and Flipped Classroom report results on these datasets and include more extensive baselines.
***

- SelEX is no longer the current state of the art on fine-grained datasets as this paper claims since there are newer works in GCD [A,B,C,D] with [A,B,C] all demonstrating to outperform Selex. Therefore framing comparisons primarily against selex  misrepresents the true standing of the proposed method. These newer works should be added in the comparison.
***

- While the proposed approach performs well on the fine-grained datasets, outperforming all methods compared to, the results are more mixed on the generic datasets where on some metric FlipClass outperforms.
- I don't think this is a major issue as the performance is still strong, however it would have been interesting to see if better results could be achieved by combining the proposed approach and FlipClass
***

[A] Dai et al. Adaptive Part Learning for Fine-Grained Generalized Category Discovery: A Plug-and-Play Enhancement. CVPR 2025.
***
[B] Liu et al. Hyperbolic Category Discovery. CVPR 2025.
***
[C] Tang et al. Dissecting Generalized Category Discovery: Multiplex Consensus under Self-Deconstruction. ICCV 2025.
***
[D] Xu et al. A Hidden Stumbling Block in Generalized Category Discovery: Distracted Attention. ICCV 2025.

**Questions:**

- Comparison to [A]: Could the methodological and numerical difference to [A] be clarified?
***

- Necessity of parts: Could the reported improvements be explained by background suppression rather than true part reasoning? Have you tested a variant using only a single (foreground) part to isolate this effect?
***

- More challenging datasets: Why are other baselines (e.g., Flipped Classroom, SelEX) omitted from the comparisons on Herbarium and Oxford Pets, given that results for these datasets are already publicly available?
***

- Further comparisons: Why are more recent methods e.g. [A,B,C,D] omitted from the state-of-the-art comparison?
***

- Understanding features: DINOv2 consistently outperforms DINOv3 in your experiments. Can you provide an explanation or hypothesis for this surprising trend?

---

> ### Author Response · Authors · 2025-11-21
> **Rebuttal by Authors for R4 (#rMXF) (Response 1/2)**
>
> #### We sincerely thank **R4 (#rMXF)** for the positive evaluation and thoughtful comments. We are glad the paper is well presented and appreciate the recognition of our clear motivation, simple and effective framework, and clear writing. We also thank R4 for highlighting our DINOv3 results, which, to the best of our knowledge, are the first to evaluate GCD on DINOv3. This feedback affirms the relevance and impact of our work. Below, we address the concerns point by point.
>
> - #### **R4.1 - Difference between APL and PartCo.**
>   #### **Response:** APL is a concurrent work that also exploits part-level cues for GCD, but its formulation and behavior differ substantially from ours. APL learns a set of global, learnable part queries guided by DINO priors and an all-min part contrastive loss, and then replaces the backbone CLS feature with an aggregated part representation inside a given GCD framework. In contrast, PartCo introduces explicit part-level correspondence labels derived from frozen ViT patch tokens (via PCA and clustering) and uses them as regularization with generic part-level correspondence loss; we do not add learnable queries, do not modify the classifier head, and introduce no extra inference cost, making PartCo a truly plug-and-play module for both parametric and non-parametric GCD backbones (SimGCD, SelEx, SPTNet, FlipClass, etc.). On shared fine-grained benchmarks, integrating our explicit correspondence labels and loss yields consistently and significantly strong results, indicating that PartCo effectively leverages part-level cues for category discovery. We added a short explanation in the Related Work section (Supplementary, Section S1) and the experimental comparison with APL in the updated the **Main paper, Table 1** to clearly articulate these conceptual and empirical differences.
>
> - #### **R4.2 - On the necessity of explicit part modeling.**
>   #### **Response:** We thank the reviewer for this insightful question. To directly address it, we added an ablation where we simply mask background regions and train the baselines using foreground-only features, without any explicit part correspondence (see **Supplementary, Section S3.6**). For SimGCD on CUB and Stanford Cars, this foreground-only variant yields only modest gains over the original baseline (e.g., CUB: 71.5→73.7 “All”, Cars: 71.5→73.0 “All”), indicating that removing background alone provides limited benefit. In contrast, introducing explicit part-level correspondence with PartCo leads to substantially larger improvements on the same backbones and datasets (see the **updated Main paper, Table 1 & 2**), showing that the advantage does not come merely from suppressing background but from explicitly aligning semantically meaningful object parts across images. We will incorporate this ablation and discussion into the revised version to clarify why explicit part modelling is necessary and beneficial.
>
> - #### **R4.3 - More experiment on Herbarium & Pets datasets**
>   #### **Response:** We appreciate the reviewer’s suggestion to include stronger baselines on more challenging, long-tailed datasets. While we already report results on Herbarium and Oxford Pets with SimGCD in the supplementary material, we agree that adding SelEx and FlipClass (and their PartCo variants) would give a more complete picture. Due to the limited time available in the rebuttal period and the higher computational cost of these long-tailed settings, we were not able to finish all additional runs for these datasets in time. However, we are in the process of extending our experiments to include SelEx and FlipClass on Herbarium and Pets, and we will incorporate these results and a corresponding discussion in the revised version of the paper.
>
> - #### **R4.4 - Comparison with recent works.**
>   #### **Response:** We thank the reviewer for pointing this out and agree that comparisons should reflect the current state of the field. In the revised version, we have expanded our experimental comparison to include recent GCD methods beyond SelEx. Specifically, we add results from works recently published in ICLR’25 [a], CVPR’25 [b, c, d], ICCV’25 [e, f, g], TMM [h], ICML’25 [i], and NeurIPS’25 [j, k], as summarized in the **updated Main paper, Table 1** for fine-grained SSB benchmarks and **Table 2** for generic benchmarks. Across these settings, our method consistently outperforms all compared approaches (and is on par with Hyp-SelEx [d] when compared in the PartCo-SelEx configuration), while significantly improving each backbone it is applied to. These results clarify that our contributions remain competitive against the latest SOTA and further confirm the effectiveness and robustness of our part-level correspondence prior framework.

---

> ### Author Response · Authors · 2025-11-21
> **Rebuttal by Authors for R4 (#rMXF) (Response 2/2)**
>
> - #### **R4.5 - Improving FlipClass with PartCo.**
>   #### **Response:** As shown in the **updated Main paper, Table 1**, we show that PartCo indeed improve FlipClass in many challenging datasets. This further confirms the effectiveness and robustness of our part-level correspondence prior framework.
>
> - #### **R4.6 - On the DINOv3 results.**
>   #### **Response:** Thank you for the thoughtful comment. The leading hypothesis is suboptimal fine-tuning for DINOv3. In this work, DINOv3 was fine-tuned with a DINOv2-style recipe (learning rate, decay, augmentations, layer-wise LR, etc.). Given differences in architecture and pretraining, DINOv3 likely needs a tailored setup. As future work, we will study DINOv3 more in depth (LR/warmup, weight decay, augmentation strength, layer-wise decay, head tuning, resolution, mixup/cutmix). Current results are best viewed as a lower bound for DINOv3 under a DINOv2-like protocol.
>
> #### **Note:** Full citation details for [a–k] are provided in the [General Response](https://openreview.net/forum?id=xiKtlg1jED&noteId=YrcQnibIYn).

---

> > ### Comment · Reviewer_rMXF · 2025-11-27
> >
> > Thank you to the authors for their detailed reply to my comments. I appreciate the clarification on the difference to APL, the experiment on background removal vs. part modelling and the additional comparisons to related works including APL and the combination of the proposed method with FLIPClass, further demonstrating its ability to plug into and improve existing models.
> >
> > The majority of my concerns have been addressed by this response, the only remaining concern is the more competitive baselines on Herbarium and Oxford Pets. I understand why the authors wouldn't have time to run these experiments as they already provided a lot of new experiments in the rebuttal and while Selex and Flipped Classroom do provide results on Herbarium they use DINO features rather than the DINOv2 features used for Herbarium dataset in this paper. This is not a major concern as the performance over SimGCD is considerable here and the paper has already demonstrated excellent performance on the 6 commonly used datasets for GCD in Tables 1 and 2. I therefore have raised my rating.

---

> > > ### Author Response · Authors · 2025-11-27
> > > **Official Comment of Submission10874 by Authors**
> > >
> > > #### Dear R4 (#rMXF),
> > >
> > > #### Thank you for your thoughtful follow-up and for raising your rating. We appreciate your careful assessment and the recognition of our additional experiments and clarifications.
> > >
> > > #### Regarding the other baselines on Herbarium and Oxford Pets: we are currently running these experiments and will include the results in the camera-ready version. We will also post an update in the discussion as soon as the results are ready.
> > >
> > > #### Thank you again for your constructive feedback and support.
> > >
> > > #### Best regards,
> > > #### Author of Submission 10874

---

> > > > ### Author Response · Authors · 2025-12-04
> > > > **Official Comment of Submission10874 by Authors**
> > > >
> > > > #### Dear R4 (#rMXF),
> > > >
> > > > #### Thank you again for your constructive & positive feedback. We truly appreciate your thoughtful assessment.
> > > >
> > > > #### As a follow-up to our earlier note, we’ve completed the additional experiments on Oxford Pets and Herbarium 19 for three of our baselines (SimGCD, FlipClass, and SelEx). Across both datasets, PartCo consistently enhances all baselines and achieves overall better results. The detailed numbers are provided in the **Supplementary, Table C**.
> > > >
> > > > #### We will also reflect these results in the camera-ready version. Thank you once more for your time, constructive comments, and support.
> > > >
> > > > #### Best regards,
> > > > #### Author of Submission 10874

---

### Official Review · Reviewer_fJDg · 2025-10-26

**Soundness:** 3
**Presentation:** 3
**Contribution:** 3
**Rating:** 4
**Confidence:** 4

**Summary:**

This paper introduces PartCo, a framework that enhances Generalized Category Discovery (GCD) by incorporating part-level correspondence priors extracted from ViT patch tokens.

**Strengths:**

- Clear motivation for why part-level information helps GCD, especially for fine-grained categories with shared global features but different local compositions
- Simple and effective approach that leverages existing foundation models without requiring additional annotations or complex architectural changes
- Genuinely plug-and-play—demonstrated compatibility with multiple GCD baselines (SimGCD, SelEx, SPTNet, etc.) with consistent improvements

**Weaknesses:**

- The novelty is somewhat limited. Using patch tokens from foundation models to extract fine-grained features is increasingly common. The specific formulation of first and second-order correspondences is the main technical contribution, but this feels incremental.

- The reliance on DINO features for correspondence labels means PartCo inherits DINO's biases and limitations. If DINO fails to capture relevant part correspondences (e.g., due to severe occlusion or unusual viewpoints), PartCo will struggle. This dependency is acknowledged in limitations but not thoroughly analyzed.

- The choice of second-order correspondence is not well-justified theoretically. Why stop at second-order? Have you tried third-order or higher? Is there a principle for selecting the order, or is it empirical?

- Limited analysis of failure cases or when part-level features don't help. The paper shows PartCo improves performance on average, but are there categories or scenarios where it hurts? For example, what about categories distinguished by global shape rather than local parts?

- Computational cost analysis is incomplete. How much does PartCo add to training time and memory? The paper mentions it's "lightweight" but doesn't provide concrete numbers or scalability analysis.

- The part correspondence labels are treated as pseudo-ground-truth, but their quality is not validated. How accurate are these correspondences? Some analysis comparing against manually annotated part correspondences (available in CUB) would strengthen the paper.

- The paper claims part-level features help with novel category discovery, but the analysis mostly focuses on fine-grained datasets where this is intuitive. The gains on ImageNet and CIFAR are smaller, suggesting limited benefit for coarse-grained categories. This limitation deserves more discussion.

**Questions:**

1. Have you validated the quality of correspondence labels against ground-truth part annotations in CUB? What's the alignment accuracy?

2. Why second-order specifically? Can you provide ablations on first-order only, second-order only, and third-order to justify this choice?

3. How does PartCo perform when the foundation model (DINO) fails to capture good part correspondences? Can you show failure cases?

4. What is the computational overhead? Please provide training time, memory usage, and FLOPs comparisons.

5. For coarse-grained categories (ImageNet, CIFAR), the gains are modest. Can you analyze which types of categories benefit from part-level features and which don't?

6. How sensitive is PartCo to the hyperparameter λp (correspondence loss weight)? The ablation shows some variation but doesn't analyze why certain values work better for different datasets.

7. The attention maps show PartCo focuses on parts, but how does this translate to better novel category discovery specifically? Can you show examples where part-level features helped correctly cluster novel categories that global features confused?

---

> ### Author Response · Authors · 2025-11-21
> **Rebuttal by Authors for R3 (#fJDg) (Response 1/2)**
>
> #### We sincerely thank **R3 (#fJDg)** for the positive evaluation and thoughtful comments. We appreciate reviewer recognition of our paper’s clear motivation and of our simple, truly plug-and-play approach that requires no additional annotations or complex architectural changes, yet delivers strong and consistent performance across multiple benchmarks. This feedback affirms the relevance and impact of our work. Below, we address the concerns point by point.
>
> - #### **R3.1 - On the novelty of the proposed methods.**
>   #### **Response:** We thank the reviewer for this comment. While we agree that using patch tokens from foundation models is becoming increasingly common, we believe that the *specific idea and realization* we propose have not been studied in the GCD literature. We also feel it is important to assess novelty in the right context and problem setting: GCD differs fundamentally from standard classification or fine-grained recognition, as it must discover both known and novel categories in an unlabeled pool given only partial supervision.
>   #### In this work, we introduce *explicit part-level correspondence learning* for GCD and instantiate it in a tailored framework, PartCo, that facilitates knowledge transfer from seen to unseen categories. This is motivated by a key insight in GCD [Vaze et al., 2022; Wang et al., 2024] that object parts are effective vehicles for transferring knowledge from known to novel classes. To the best of our knowledge, no prior GCD method provides an explicit mechanism to operationalize this insight; our work is, as far as we are aware, the first to do so in a way that is compatible with a wide range of GCD backbones. Specifically: **(1)** Explicit part-level correspondence labels: PartCo assigns and aligns shared part labels across images, offering a fine-grained, semantically grounded inductive bias that guides what local structures are transferable, something existing GCD methods with global features cannot explicitly leverage. **(2)** Part-level correspondence loss: We introduce a loss that pulls embeddings of regions with the same part label/category together and pushes different parts apart, enabling robust part-aware features that directly support transfer to unlabeled novel classes.
>   #### No prior GCD method supports explicit part-level correspondence learning. Our ablations and results show this simple, effective inductive bias consistently improves performance and achieves SOTA across multiple GCD benchmarks, while remaining plug-and-play with various backbones (updated **Main paper, Tables 1–2; Supplementary, S3.4**).
>
> - #### **R3.2 - Concerns regarding the relience on DINO features.**
>   #### **Response:** We thank the reviewer for highlighting this important point. We would first like to clarify that, in our limitations section, we do not state that PartCo is specifically limited by DINO itself. Rather, the current framework is built on top of transformer-based architectures and therefore relies on access to patch tokens to compute part-level correspondences. In principle, any ViT-style foundation model that provides patch embeddings can be used to instantiate PartCo.
>   #### To examine this more thoroughly and assess generalization beyond DINO, we conducted additional experiments using CLIP’s vision encoder (Table R3-1). Even when we only use the CLIP image encoder, PartCo still consistently and significantly improves the underlying GCD baselines across multiple datasets, achieving strong overall performance. This suggests that our framework does not hinge on a particular pre-training recipe (e.g., DINO) but instead leverages a more general property of modern vision transformers: their ability to provide reasonably structured patch-level features.
>   #### We agree that, as with any method that builds on a frozen backbone, PartCo will inherit some biases and failure modes of the underlying foundation model (e.g., under severe occlusion or extreme viewpoints). However, the empirical results with both DINOv2/v3 and CLIP indicate that PartCo is robust across different pre-trained transformers, and that its benefits are not confined to a single pre-trained family.

---

> > ### Author Response · Authors · 2025-11-21
> > **Rebuttal by Authors for R3 (#fJDg) (Response 2/2)**
> >
> > - #### **R3.3 - Regarding the choice of PartCo's labels.**
> >   #### **Response:** We appreciate the concern and have extended our study to explicitly evaluate higher-order correspondences. The revised paper now includes an analysis of third-order part correspondences (**Supplementary, Section S3.5**). Qualitatively (see **supplementary, Figure A**), extending PartCo to third order over-fragments parts: the same semantic part (e.g., a bird’s head or wing) splits into multiple labels, yielding redundant, noisy correspondences that are less meaningful and harder to transfer. Quantitatively (**Supplementary, Table I**), training with third-order labels consistently degrades performance across datasets compared to our 1st+2nd-order design, indicating that higher-order labels add more noise than signal.
> >   #### Taken together, these results suggest a practical principle: 1st order and 2nd order correspondences strike the right balance, capturing part identity and pairwise co-composition for both fine-grained and generic settings, whereas higher orders over-cluster and leads to ineffective part corresponding learning.
> >
> > - #### **R3.4 - Computational overhead analysis.**
> >   #### **Response:** We report concrete training and compute statistics in **Supplementary, Section S2.3**. Compared to SimGCD, PartCo-SimGCD incurs only modest overhead on CUB / Cars / CIFAR100: training time increases by just 5%, 2.5%, and 4.2%, peak memory rises from ~8.6 GB to ~12.0 GB, and GFLOPs grow from 54.04 G to 62.58 G (≈+15%). This extra cost mainly comes from computing patch-level correspondences and the additional projection head. Importantly, inference remains unchanged: at test time, PartCo uses the same backbone features and clustering pipeline as the underlying GCD method, so there is no additional inference overhead. We believe this small training-time cost is well justified by the consistent significant accuracy gains observed across benchmarks.
> >
> > - #### **R3.5 - On the quality and effectiveness of PartCo labels.**
> >   #### **Response:** To verify the effectiveness and quality of our part-level correspondence labels, we compare our constructed CUB’s part correspondence labels (before downsample) against the manually annotated parts provided by CUB dataset and report the results in the **Supplementary, Table. H**. For each visible ground-truth part, we check whether the assigned part labels matches the corresponding CUB part through hungarian matching algorithm. The results show that our 1st and 2nd order part-level correspondence labels achieve overall accuracies of 80.8% and 77.5%, respectively, with especially strong alignment on key semantics parts such as *back, beak, belly, leg, wing, and tail*. Moreover, the significant performance gains reported in the **Main paper, Table. 1 and 2**, suggest that the part-level correspondence labels are of sufficiently high quality to provide a reliable supervisory signal for our part-level correspondence learning framework.
> >
> > - #### **R3.6 -  On limited gains in generic benchmarks.**
> >   #### **Response:** We agree with the reviewer that the improvements on generic datasets (e.g., ImageNet, CIFAR) are smaller than on fine-grained ones. We believe there are two main reasons. **(1)** as the reviewer notes, coarse-grained benchmarks often already admit highly discriminative global representations; in such settings, part-level correspondence provides a complementary but less dominant signal, so the absolute gains are naturally smaller. Importantly, we still observe consistent improvements, because **PartCo acts as an additional regularizing objective on patch-level features:** it does not overwrite the global shape information learned by the backbone, but rather refines it when reliable part correspondences are available. **(2)**, these generic benchmarks are already close to saturation for strong GCD methods; despite this, our approach reaches very high accuracy (e.g., 94.6% average), achieving **state-of-the-art** performance compared with recent GCD methods (**updated Main paper, Table 1 & 2**).
> >
> > - #### **R3.7 - Sensitivity of PartCo's balancing weight.**
> >   #### **Response:** A balancing factor λb is introduced to regulate the balance between our proposed loss and the baseline’s losses. In search of the optimal balancing factor, we investigate a range of weight values, ranging from 0, 0.1, 0.35, 0.7, and 1.0. The results, as shown in the **Main paper, Table 4**, demonstrate that a fixed weight 0.35 is robust and consistently achieve the highest accuracy gains, highlighting PartCo’s strong practical generalization.

---

> > > ### Comment · Reviewer_fJDg · 2025-11-25
> > >
> > > Thank you for your response. I appreciate the additional experiments with CLIP backbone, the third-order correspondence analysis, and the part-label quality validation against CUB annotations. However, specific failure case analysis showing when part-level features hurt or don't help remains limited. I will maintain my initial score.

---

> > > > ### Author Response · Authors · 2025-11-26
> > > > **Additional Response to R3 (#fJDg) on PartCo's Success & Failure Cases**
> > > >
> > > > #### We thank the reviewer for the follow-up and for acknowledging our additional experiments (CLIP backbone, third-order correspondence, and part-label validation). Following R3 requests for a more in-depth analysis of when part-level features help or hurt, we have added a dedicated success/failure study in the **Supplementary, Section S5**.
> > > >
> > > > - #### **R3.8 - Success & failure analysis.**
> > > >   #### **Response:** In **Tab. K**, we report per-class accuracy differences between PartCo and the baseline on CUB and highlight the Top 10 largest gains and Top 10 largest drops, with qualitative visualizations in **Fig. E** and **Fig. F**.
> > > >   #### The analysis shows that PartCo is most beneficial for fine-grained, part-dependent species with similar global shape but distinctive local plumage patterns, whereas the main failures correspond to classes where global shape is more informative than any single local patch, or where local patches are visually ambiguous or highly heterogeneous (e.g., juvenile vs. adult appearance, see **Fig. G**). Importantly, the largest gains (up to **+100 points**, often on unseen classes) are considerably larger than the largest drops, supporting our claim that part-level cues are an effective vehicle for transferring fine-grained discriminative knowledge from seen to unseen categories.
> > > >
> > > > #### It is hoped that this additional analysis addresses the raised concerns and motivates R3's reconsideration of the overall assessment. Please feel free to let us know if there are any further questions or points that would benefit from clarification.

---

### Official Review · Reviewer_nFJh · 2025-11-01

**Soundness:** 2
**Presentation:** 3
**Contribution:** 2
**Rating:** 4
**Confidence:** 4

**Summary:**

This paper presents a new learning framework for Generalized Category Discovery (GCD) by integrating explicit part-level visual feature correspondences. In contrast to traditional GCD methods, the proposed approach enhances category understanding and discovery by leveraging compositional object features. Experimental results on benchmark datasets demonstrate that the method achieves improved performance in most cases.

**Strengths:**

1. The motivation of the paper is clear, and the content is easy to understand.
2. The idea of incorporating explicit part-level visual feature correspondences is interesting.
3. The experimental results indicate that the proposed method outperforms existing approaches across different benchmark datasets in most scenarios.

**Weaknesses:**

1. The technical novelty appears somewhat limited, particularly as the work is built upon the existing SimGCD framework.
2. The generalization capability of the method has not been fully validated, as only two baseline methods are tested. Evaluation with more benchmark methods is recommended.
3. Most existing GCD methods use pre-trained models different from DINOv2 and DINOv3, making it difficult to fairly compare the results with those reported in original studies. It is suggested to maintain consistency in the pre-trained model for a more equitable comparison.

**Questions:**

See weaknesses

---

> ### Author Response · Authors · 2025-11-21
> **Rebuttal by Authors for R2 (#nFJh)**
>
> #### We sincerely thank **R2 (#nFJh)** for the constructive feedback and appreciate the positive recognition of our clear motivation, interesting method and strong performance. Below we address reviewer R2’s questions and concerns.
>
> - #### **R2.1 - On the novelty & generalization capability of the proposed method.**
>   #### **Response:** We thank the reviewer for raising this point and would like to clarify that our proposed framework is *not* tied to SimGCD. PartCo is explicitly designed to be model-agnostic: as shown in the **Main paper, Table 1 & 2**, it can be seamlessly integrated with a range of parametric and non-parametric GCD baselines, including SimGCD, SPTNet, FlipClass, and SelEx. Across all these backbones (DINOv2,v3 and CLIP), PartCo consistently brings substantial performance gains, which confirms that it acts as a generic, plug-and-play part correspondence prior rather than a SimGCD-specific extension.
>   #### Regarding technical novelty, our main contribution is to introduce explicit part-level correspondence learning for GCD and to instantiate it in a tailored framework (PartCo) that facilitates knowledge transfer from seen to unseen categories. This is motivated by a key insight in GCD [Vaze et al., 2022; Wang et al., 2024] that object parts provide an effective vehicle for transferring knowledge from known to novel classes. However, to the best of our knowledge, no prior GCD method **explicitly** operationalizes this insight through a dedicated part-correspondence mechanism. PartCo is the first framework that (i) formulates part correspondences labels and loss for GCD and (ii) integrates them in a manner that is both effective and easily composable with existing GCD methods.
>
> - #### **R2.2 - On the usage of DINOv2.**
>   #### **Response:** We thank the reviewer for raising this important point. Recently, there has been a surge of stronger vision foundation models, and the GCD community is increasingly standardising on DINOv2 to keep category discovery research aligned with the current state of the field, as also discussed in the recent survey [p]. Following this trend, a number of recent GCD papers have begun to reimplement or report results with DINOv2 backbones; we summarise these works in the **updated Main paper, Table 1 and 2**.
>   #### In line with this, our main comparisons adopt DINOv2 as the common backbone, and we additionally report, to the best of our knowledge, the first GCD results using DINOv3 (see Tables 1 and 2). Across both DINOv2 and DINOv3, PartCo consistently yields strong performance and clear improvements over all considered baselines.
>   #### To further demonstrate that our framework is not tied to DINO-style pre-training, we also include experiments using CLIP vision encoder as the backbone (**Supplementary, Section S3.4**). Even when relying solely on the CLIP image encoder, the resulting representations are strong, and integrating PartCo again provides consistent gains.
>   #### Overall, these experiments show that PartCo is robust and effective across different foundation models, supporting the generalization and robustness of our framework.
>
> #### **Note:** Full citation detail for [p] is provided in the [General Response](https://openreview.net/forum?id=xiKtlg1jED&noteId=YrcQnibIYn).

---

> > ### Author Response · Authors · 2025-11-27
> > **Kind Reminder for R2 (#nFJh)**
> >
> > #### Dear R2 (#nFJh),
> >
> > #### We submitted our initial rebuttal on 21 Nov 2025, where we addressed most of your questions in a structured manner and summarized the manuscript revisions in the [General Response](https://openreview.net/forum?id=xiKtlg1jED&noteId=YrcQnibIYn). As a gentle note, the author-reviewer discussion period will end in about a week. When you have a moment, could you please take a look and let us know if any points need further clarification? We’re happy to provide additional details or evidence as needed.
> >
> > #### Thank you for your time and consideration.
> >
> > #### Best regards,
> > #### Author of Submission 10874

---

### Official Review · Reviewer_bQ1m · 2025-11-03

**Soundness:** 3
**Presentation:** 3
**Contribution:** 3
**Rating:** 6
**Confidence:** 4

**Summary:**

This paper proposes a novel framework, PartCo that introduces a part-level correspondence prior to improve category discovery by leveraging fine-grained visual relationships beyond global image features. It integrates seamlessly with existing GCD methods and enhances both supervised and unsupervised learning through a novel part-level contrastive loss. Experiments across multiple benchmarks show significant accuracy gains and state-of-the-art performance, validating its effectiveness and adaptability.

**Strengths:**

(1) The proposed method, PartCo,  effectively integrates part-level visual priors into existing GCD frameworks, enhancing fine-grained category discrimination without altering model architecture.

(2) It consistently achieves state-of-the-art performance across multiple benchmark datasets, demonstrating strong robustness and generalizability.

**Weaknesses:**

Related works are not discussed properly. More recent works are needed to be cited.

(1) Adaptive Part Learning for Fine-Grained Generalized Category Discovery: A Plug-and-Play Enhancement, CVPR 2025

(2) Hyperbolic Category Discovery, CVPR 2025

(3) Cdad-net: Bridging domain gaps in generalized category discovery, CVPR 2024

(4) MOS: Modeling Object-Scene Associations in Generalized Category Discovery, CVPR 2025

**Questions:**

(1) Is the selection of between 1st- and 2nd-order labels dataset-dependent? How could you decide the best in general?

(2) What are the effects of the hyperparameters i.e. PCA threshold $\tau$ and the balancing factor $\lambda_{b}$, as these are empirically chosen?

---

> ### Author Response · Authors · 2025-11-21
> **Rebuttal by Authors for R1 (#bQ1m)**
>
> #### We sincerely thank **R1 (#bQ1m)** for thoughtful reading and appreciate the positive recognition of our novel framework and its strong performance. Below we address reviewer R1’s questions and concerns.
>
> - #### **R1.1 - Comparison with recent works.**
>   #### **Response:** We appreciate the reviewer’s valuable suggestion. To strengthen our comparison with recent work, we have added results from papers published in ICLR’25 [a], CVPR’25 [b, c, d], ICCV’25 [e, f, g], TMM [h], ICML’25 [i], and NeurIPS’25 [j, k]. These results are reported in the **updated Main paper: Table 1** (fine-grained SSB benchmarks) and **Table 2** (generic benchmarks). In the same tables, **we also include additional baselines** (e.g., SPTNet, FlipClass) and their variants integrated with our PartCo framework (PartCo-SPTNet and PartCo-FlipClass).
>   #### Overall, our method consistently outperforms all compared approaches across datasets (and is on par with Hyp-SelEx [d] when compared with PartCo-SelEx), while substantially improving each baseline. These results further confirm the effectiveness and robustness of our part-level correspondence prior. We will include these updates in the revised version to provide a more comprehensive evaluation across recent methods and pretrained backbones.
>
> - #### **R1.2 - On the selection between 1st- and 2nd-order labels.**
>    #### **Response:** The selection between 1st- and 2nd-order labels is grounded in empirical evidence. As shown in the Main paper ablation studies, **Figure 7**, 1st-order labels consistently yield stronger performance on fine-grained recognition tasks, whereas 2nd-order labels deliver better results on more generic recognition tasks.
>
> - #### **R1.3 - Effects of PartCo hyperparameters.**
>   #### **Response:** (a) PCA threshold: fixed to 0.6 for all datasets, determined empirically. Using a single threshold across benchmarks shows that PartCo’s label generation is robust and does not require any dataset-specific adaptive strategy, while maintaining a stable trade-off between dimensionality reduction and information retention. (b) Balancing factor (PartCo): we provide ablation study in the **updated Main paper, Table 4**. The results show that a fixed weight of 0.35 is robust and well balanced, highlighting strong practical generalization.
>
> #### **Note:** Full citation details for [a–k] are provided in the [General Response](https://openreview.net/forum?id=xiKtlg1jED&noteId=YrcQnibIYn).

---

> > ### Author Response · Authors · 2025-11-27
> > **Kind Reminder for R1 (#bQ1m)**
> >
> > #### Dear R1 (#bQ1m),
> >
> > #### We submitted our initial rebuttal on 21 Nov 2025, where we addressed most of your questions in a structured manner and summarized the manuscript revisions in the [General Response](https://openreview.net/forum?id=xiKtlg1jED&noteId=YrcQnibIYn). As a gentle note, the author–reviewer discussion period will end in about a week. When you have a moment, could you please take a look and let us know if any points need further clarification? We’re happy to provide additional details or evidence as needed.
> >
> > #### Thank you for your time and consideration.
> >
> > #### Best regards,
> > #### Author of Submission 10874

---

### Author Response · Authors · 2025-11-21
**General response of Submission 10874 by Authors**

#### We thank all reviewers, **R1 (#bQ1m)**, **R2 (#nFJh)**, **R3 (#fJDg)**, and **R4 (#rMXF)** for their thoughtful and constructive feedback, and in particular, for recognizing the key contributions of our well motivated work in Generalized Category Discovery (GCD). Across the reviews, there is broad agreement on the strong motivation and key insight presented in our work, i.e., **(1)** the importance of incorporating explicit part-level learning for GCD (all reviewers). **(2)** Moreover, several reviewers positively highlighted both our simple, genuinely plug-and-play, and effective framework design (R1, R3, R4), **(3)** Its strong performance gains across different baseline models and strong robustness and generalizability across datasets (all reviewers), **(4)** as well as our well-written paper and clear exposition (R2, R4).

#### We have carefully addressed all comments in the rebuttal (organized by reviewer and question index **R{reviewer}.{question}**, e.g., **R1.1**, **R3.6**, etc.), and we have revised the paper accordingly. **The supplementary material can be found in the same PDF as the main paper**, immediately following the References section. We are happy to address any remaining concerns.

#### Below, we summarize the major updates made to the paper (All revisions are marked in blue) during the discussion period:
- #### **Main paper, Tables 1 and 2:** Added comparisons with 11 recent works published in ICLR’25 [a], CVPR’25 [b, c, d], ICCV’25 [e, f, g], TMM [h], ICML’25 [i], NeurIPS’25 [j, k]), per R1 (**R1.1**), and R4 (**R4.4**). Also added new baselines SPTNet and FlipClass for PartCo (R2, **R2.1**).
- #### **Main paper, Table 4:** Added an ablation on the balancing factor, addressing R1 (**R1.3**) and R3 (**R3.7**).
- #### **Supplementary, Section S3.4:** Added experiments with a CLIP backbone [l,m,n,o] (R3, **R3.2**).
- #### **Supplementary, Section S3.5, Table I:** Added analysis on choosing 1st- vs 2nd-order correspondence labels (R3, **R3.3**).
- #### **Supplementary, Section S2.3:** Added computational overhead details (R3, **R3.4**).
- #### **Supplementary, Section S3.5, Table H:** Added analysis of part-label quality and effectiveness (R3, **R3.5**).
- #### **Supplementary, Section S1:** Expanded related work and clarified differences from concurrent works (R4, **R4.1**).
- #### **Supplementary, Section S3.6:** Justified the use of explicit part-level correspondence learning (R4, **R4.2**).

#### References
- #### [a] Liu, Yuanpei, and Kai Han. "Debgcd: Debiased learning with distribution guidance for generalized category discovery." In ICLR. 2025.

- #### [b] Dai, Qiyuan, et al. "Adaptive Part Learning for Fine-Grained Generalized Category Discovery: A Plug-and-Play Enhancement." In CVPR. 2025.

- #### [c] Peng, Zhengyuan, et al. "MOS: Modeling Object-Scene Associations in Generalized Category Discovery." In CVPR. 2025.

- #### [d] Liu, Yuanpei, Zhenqi He, and Kai Han. "Hyperbolic category discovery." In CVPR. 2025.

- #### [e] Xu, Qiyu, et al. "A hidden stumbling block in generalized category discovery: Distracted attention." In ICCV. 2025.

- #### [f] Cao, Xinzi, et al. "AllGCD: Leveraging All Unlabeled Data for Generalized Category Discovery." In ICCV. 2025.

- #### [g] Tang, Luyao, et al. "Dissecting generalized category discovery: Multiplex consensus under self-deconstruction." In ICCV. 2025.

- #### [h] Wang, Enguang, et al. "Learning Part Knowledge to Facilitate Category Understanding for Fine-Grained Generalized Category Discovery." IEEE Transactions on Multimedia. 2025.

- #### [i] Liu, Duo, et al. "Generalized Category Discovery via Reciprocal Learning and Class-Wise Distribution Regularization." In ICML. 2025.

- ##### [j] He, Zhenqi, Yuanpei Liu, and Kai Han. "Seal: Semantic-aware hierarchical learning for generalized category discovery." In NeurIPS. 2025.

- #### [k] Han, Jizhou, et al. "Unleashing the Power of Neural Collapse: Consistent Supervised-Unsupervised Alignment for Generalized Category Discovery." In NeurIPS. 2025.

- #### [l] Zheng, Haiyang, et al. "Textual knowledge matters: Cross-modality co-teaching for generalized visual class discovery." In ECCV. 2024.

- #### [m] Wang, Enguang, et al. "Get: Unlocking the multi-modal potential of clip for generalized category discovery." In CVPR. 2025.

- #### [n] Ouldnoughi, Rabah, Chia-Wen Kuo, and Zsolt Kira. "Clip-gcd: Simple language guided generalized category discovery." arXiv preprint arXiv:2305.10420. 2023.

- #### [o] Yang, Muli, et al. "Consistent prompt tuning for generalized category discovery." International Journal of Computer Vision. 2025.

- #### [p] He, Zhenqi, Yuanpei Liu, and Kai Han. "Category Discovery: An Open-World Perspective." arXiv preprint arXiv:2509.22542. 2025.

---

> ### Author Response · Authors · 2025-11-26
> **Additional General response of Submission 10874 by Authors**
>
> #### In response to R3’s concerns about PartCo’s failure cases, the manuscript has been further updated as follows:
> - #### **Supplementary, Section S5, Table K:** Added a detailed success and failure case analysis for PartCo, including per-class accuracy deltas and qualitative visualizations on CUB, clarifying when part-level features help or hurt (R3, **R3.8**).

---

### Author Response · Authors · 2025-12-01
**Author Final Remarks of Submission 10874 by Authors**

#### Dear Area Chair,
#### The authors are very grateful to the Area Chair and Reviewers for their time, constructive feedback and discussion, which has made the paper significantly stronger overall. After the rebuttal and revisions, the overall assessment is positive, with all of the concerns having been addressed. Below, each point explicitly links what reviewers requested to what has been provided in the revised submission.

- #### **R1 (#bQ1m)**
  > #### **What R1 asked for:** (i) Stronger positioning against recent GCD works; (ii) clarification on selecting between 1st- and 2nd-order labels; (iii) analysis of key hyperparameters (PCA threshold and balancing factor).
> #### **Response:** The revised paper (a) adds results for 11 recent methods [a-p] (ICLR’25, CVPR’25, ICCV’25, ICML’25, NeurIPS’25, TMM) in the **updated Main Paper, Tab. 1-2**, along with new baselines (SPTNet, FlipClass and their PartCo variants), showing that PartCo consistently brings significant gains across datasets and backbones; (b) explains the empirical choice of 1st vs. 2nd order labels (**Fig. 7** & **Supplementary Sec. S3.5**); and (c) provides an ablation on the balancing factor λb (**Tab. 4**), demonstrating robust behaviour without dataset-specific tuning.

- #### **R2 (#nFJh)**
  > #### **What R2 asked for:** Clarification on whether the method is essentially a SimGCD extension, and clearer evidence of novelty and generalisation beyond the original baselines and specific pretraining choices.
> #### **Response:** The rebuttal and revision (a) clarify that PartCo is explicitly designed to be model-agnostic, and now report results with multiple additional backbones (SimGCD, SPTNet, FlipClass, SelEx, APL) and foundation models (DINOv2, DINOv3, CLIP) in updated **Main Tables 1-2** and **Supplementary Sec. S3.4** respectively; and (b) sharpen the statement of the core novelty: explicit part-level correspondence priors for GCD (label construction and part-correspondence loss) that transfer fine-grained structure from seen to unseen categories while remaining plug-and-play. These additions directly address concerns that the method is merely a SimGCD variant and that its generalisation was under-validated.

- #### **R3 (#fJDg)**
   > #### **What R3 asked for:** R3’s comments were mostly requests for clarification and deeper understanding of the method, e.g., reliance on DINO features, the choice of PartCo labels (1st/2nd vs. higher order), computational overhead, the quality of the pseudo labels, and sensitivity to the balancing weight λb.
> #### **Response:** These points were all addressed in the first rebuttal via CLIP/foundation-model experiments (**Supplementary, Sec. S3.4**), PartCo's part-label validation (**Sec. S3.5 & Tab. H**), detailed compute analysis (**Sec. S2.3**), and a λb ablation (**Tab. 4**), which R3 explicitly acknowledged and appreciated. **R3 then further asked for one further item: a more detailed failure-case analysis**. The revised version now includes a comprehensive success/failure study (**Sec. S5, Tab. K, Figs. E-G**), showing where PartCo helps most (fine-grained, part-dependent classes) and where it can underperform. Due to an unforeseen circumstances, they were unable to continue in the discussion or update their review, but all of their requests, including the final failure-case analysis, are now addressed in the manuscript.

- #### **R4 (#rMXF)**
  > #### **What R4 asked for:** (i) Clearer description and empirical comparison to APL [a]; (ii) evidence that explicit part modelling is necessary beyond foreground masking; (iii) more up-to-date and broader comparisons, including FlipClass and recent SOTA GCD works.
> #### **Response:** The revised version adds a conceptual and numerical comparison to APL (**Supplementary Sec. S1** & **updated Main Paper, Tab. 1**); includes a foreground-only ablation showing that simple background removal yields only modest gains, while explicit part correspondences bring substantially larger improvements (**Sec. S3.6**); and expands comparisons to 11 recent GCD methods and additional baselines such as SPTNet and FlipClass, including their PartCo variants (**Tab. 1-2**). **R4’s follow-up comment explicitly states that “the majority of my concerns have been addressed”**.

#### Overall, the final paper presents a novel, truly plug-and-play part-level correspondence framework that consistently improves a wide range of GCD methods across datasets and pretraining models, with reviewers largely recognising its merit, strong motivation, strong performance, and comprehensive analysis. The detailed additional experiments, ablations, and **clarifications referred to above are summarised in the** [General Response section](https://openreview.net/forum?id=xiKtlg1jED&noteId=YrcQnibIYn) of the rebuttal and **marked in blue in the revised manuscript**.
#### Best regards,
#### Authors of Submission 10874

---

### Meta-Review · Area_Chair_rKoQ · 2025-12-29

**Summary:**

The paper received scores of 6,4,4,4.  Initial concerns include comparison to more recent GCD works, clarification with approach design choices, more analysis of key hyperparameters, novelty, and analysis of  examples where part-level features helped.

**Reviewer Concerns:**

Most of the initial concerns were adequately addressed by the rebuttal.  However, the main remaining concern is in the comparison with recent SoTA baselines.  The paper initially omitted many works over the past year, and this was flagged by multiple reviewers.  The rebuttal included many papers from 2025, and while the proposed method outperforms most of them, there is a concern.  In particular, in Table 1, Hyp-SelEx is consistently equal or better in performance than PartCo-SelEx.  Similar trends can be found in Table 2 when comparing Hyp-SimGCD and PartCo-SimGCD.  This contrasts with the rebuttal claim that "Overall, our method consistently outperforms all compared approaches across datasets (and is on par with Hyp-SelEx [d] when compared with PartCo-SelEx)". The rebuttal does not clearly explain these performance gaps with Hyp, and the wording somewhat overstates the comparative results.  As it stands, this unresolved issue weakens the strength of the empirical claims, and makes acceptance difficult to justify.

**Reviewer Scores:**

Reviewer bQ1m may have kept or lowered their score in light of the new results, as they specifically asked for the comparison to Hyp. Reviewer nFJh may have kept their score for a similar reason.  Reviewer fJDg may have kept their score or increased it as the rebuttal largely addressed their concerns.  Similarly, Reviewer rMXF may have increased their score.  So final scores might have been around two 6s and two 4s or three 6s and one 4.

---

### Decision · Program_Chairs · 2026-01-26

Reject